# Cerebellar Kv3.3 potassium channels activate TANK-binding kinase 1 to regulate trafficking of the cell survival protein Hax-1

Yalan Zhang[1], Luis Varela[2], Klara Szigeti-Buck[2], Adam Williams[3,10], Milan Stoiljkovic[2], Matija Šestan-Peša [2], Jorge Henao-Mejia[3,11], Pasquale D'Acunzo [4,5], Efrat Levy[4,5,6,7], Richard A. Flavell [3,8], Tamas L. Horvath [2] & Leonard K. Kaczmarek [1,9✉]

Mutations in *KCNC3*, which encodes the Kv3.3 potassium channel, cause degeneration of the cerebellum, but exactly how the activity of an ion channel is linked to the survival of cerebellar neurons is not understood. Here, we report that Kv3.3 channels bind and stimulate Tank Binding Kinase 1 (TBK1), an enzyme that controls trafficking of membrane proteins into multivesicular bodies, and that this stimulation is greatly increased by a disease-causing Kv3.3 mutation. TBK1 activity is required for the binding of Kv3.3 to its auxiliary subunit Hax-1, which prevents channel inactivation with depolarization. Hax-1 is also an anti-apoptotic protein required for survival of cerebellar neurons. Overactivation of TBK1 by the mutant channel leads to the loss of Hax-1 by its accumulation in multivesicular bodies and lysosomes, and also stimulates exosome release from neurons. This process is coupled to activation of caspases and increased cell death. Our studies indicate that Kv3.3 channels are directly coupled to TBK1-dependent biochemical pathways that determine the trafficking of cellular constituents and neuronal survival.

[1] Department of Pharmacology, Yale University School of Medicine, New Haven, CT, USA. [2] Department of Comparative Medicine, Yale University School of Medicine, New Haven, CT, USA. [3] Department of Immunobiology, Yale University School of Medicine, New Haven, CT, USA. [4] Center for Dementia Research, The Nathan S. Kline Institute for Psychiatric Research, Orangeburg, NY, USA. [5] Department of Psychiatry, New York University School of Medicine, New York, NY, USA. [6] NYU Neuroscience Institute, New York University School of Medicine, New York, NY, USA. [7] Department of Biochemistry & Molecular Pharmacology, New York University School of Medicine, New York, NY, USA. [8] Howard Hughes Medical Institute, Chevy Chase, MD, USA. [9] Department of Cellular and Molecular Physiology, Yale School of Medicine, New Haven, CT, USA. [10]Present address: The Jackson Laboratory for Genomic Medicine, The Jackson Laboratory, Farmington, CT, USA. [11]Present address: Department of Pathology and Laboratory Medicine, Institute for Immunology, Perelman School of Medicine, University of Pennsylvania, Children's Hospital of Philadelphia, Philadelphia, PA, USA. ✉email: leonard.kaczmarek@yale.edu

Members of the Kv3 subfamily of voltage-dependent K$^+$ channels are typically expressed in neurons that are capable of firing at high rates[1]. The Kv3.3 member of this subfamily is found at particularly high levels in Purkinje cells in the cerebellum and in many auditory brainstem nuclei. Mutations in *KCNC3*, the gene that encodes Kv3.3, result in spinocerebellar ataxia type 13 (SCA13), a human autosomal dominant disease, characterized by degeneration of the cerebellum and deficits in the processing of auditory information[2–9]. Depending on the mutation, the disease can have either an early onset, with degeneration evident within weeks of birth, or a late onset, typically in middle age.

Several experimental approaches, including a yeast-two-hybrid screen, have shown that the cytoplasmic C-terminal domain of Kv3.3 directly binds the antiapoptotic molecule Hax-1[10], a protein that is required for the normal survival of cerebellar neurons[11,12]. Hax-1 also acts as a scaffold for Rac and cortactin[13], and activates actin nucleation through the Arp2/3 complex. Thus, the interaction of Kv3.3 with Hax-1 leads to formation of a stable subcortical actin cytoskeleton under the plasma membrane where Kv3.3 is expressed. The binding of Hax-1 to Kv3.3 is also required to prevent the channel from inactivating rapidly during sustained depolarization[10].

The turnover of plasma membrane proteins such as ion channels results from their endocytosis into the endosomal pathway. This leads to the formation of late endosomes, also termed multivesicular bodies, which can then fuse with lysosomes, resulting in the degradation of the channels and their binding partners[14]. Multivesicular bodies can also fuse with the plasma membrane and expel their content into the extracellular space in the form of exosomes[15]. Cells also release intracellular material through an alternative route that encompasses the outward protrusions of portions of the plasma membrane and the direct budding of extracellular vesicles (EVs) that are named microvesicles. Microvesicles and exosomes differ in their dimensions, functions, biogenetic mechanisms, markers, and content and may potentially have different roles in neurodegenerative diseases[16].

In the present study, we have found that Kv3.3 channels form a complex with Tank Binding Kinase 1 (TBK1) and, when activated, stimulate its enzyme activity. TBK1 is a serine/threonine-protein kinase that plays a key role in autophagy and mitophagy[17,18]. TBK1 also contributes to signaling pathways of the NF$_K$B complex and is linked to diseases such as amyotrophic lateral sclerosis (ALS)[19,20]. We find that TBK1 regulates the ability of Kv3.3 to bind its ancillary subunit Hax-1. Moreover, a Kv3.3 mutant channel that is associated with late-onset ataxia causes excessive TBK1 activation both in cell lines and intact animals, enhances the trafficking of the survival protein Hax-1 into multivesicular bodies and lysosomes, stimulates the production of exosomes and increases rates of cell death. Our findings, therefore, identify a role for TBK1 in neurodegeneration caused by the G592R Kv3.3 channel.

## Results

**The G592R Kv3.1 mutation alters cerebellar function.** Previous work has shown that a human Kv3.3 mutant channel (G592R Kv3.3), which is linked to late-onset spinocerebellar ataxia, binds the cell survival protein Hax-1. Unlike wild-type channels, however, the binding of Hax-1 to the mutant channels fails to trigger the nucleation of actin filaments[10]. To probe further how an abnormal interaction between Kv3.3 and Hax-1 leads to cerebellar degeneration, we used CRISPR/Cas9 gene editing to generate homozygous G592R Kv3.3 knock-in mice (Supplementary Fig. S1). We first characterized the overall effects of this mutation on the general structure of the cerebellum and on motor behaviors. The structure of the cerebellum appeared normal by visual inspection. Using a modified version of the isotropic fractionator method to count

nuclei in frozen brain tissue[21,22], we determined that the number of cerebellar neurons was slightly reduced in the mutant mice (Fig. 1a). We found that the motor behavior of G592R Kv3.3 animals measured at ages 4 months and 7 months was significantly impaired on rotarod tests, suggesting that, as in humans, motility is impaired after middle age (Fig. 1b). As a further test for motor functions, we carried out the dowel walking test, which measures the ability of the animals to balance on a fixed wooden bar[23,24]. Again, the ability of 7-month-old G592R Kv3.3 animals was impaired when compared to wild-type animals (Fig. 1c).

EEG recordings of local field potentials across the cerebellar vermis of wild type and G592R mice indicate the mutation alters the activity of Purkinje neurons. Using a 16-channel probe, we found a significant increase in the power of spontaneous very high-frequency gamma-band oscillations (GBO, 80–300 Hz) and in the β range, while power in other frequency bands was unchanged (Fig. 1d–j). The sink-source distribution profile of GBO shows that higher power is relegated to ~180 Hz in G592R mice, while it is distributed across a wider frequency range in the wild-type mice. Furthermore, the peak of power is localized in more superficial sites of cerebellar cortex in G592R mice than in their WT counterparts (Fig.1d, heat maps), which likely corresponds to the Purkinje cell layer. These high-frequency oscillations have previously been shown to reflect the activity of recurrent inhibitory connections between Purkinje neurons[25,26], suggesting that the activity of these neurons is abnormal in the G592R animals. Increased fast oscillation rhythmicity of Purkinje cells has also been reported in another mouse model of ataxia[27].

We next compared the distribution of Kv3.3 channels in the G592R Kv3.3 animals to those in wild-type mice. By western blotting, we found that the channel protein is expressed in both the mutant and wild-type mice (Fig. 1k). At the light microscope level, immunostaining for Kv3.3 in cerebellum demonstrated that, as in wild-type animals, the G592R Kv3.3 channel was strongly localized to the somata of Purkinje neurons (Fig. 1l). To examine the subcellular localization of the channel in wild type and the G592R Kv3.3 knock-in mice, we carried out immuno-electron microscopy (immunoEM) on Purkinje neurons using diaminobenzidine labeling to localize Kv3.3 immunoreactivity. Low power EM images revealed that the channel is localized at the plasma membrane in both wild-type and mutant animals (Fig. 1m). Although the channel appeared to be expressed relatively uniformly across the somatic membrane, higher power images showed that particularly high levels of Kv3.3 exist at sites where mitochondria are closely apposed to the plasma membrane (Fig. 1n). In neurons from wild-type animals, occasional small Kv3.3-immunoreactive protrusions could be seen jutting out of the soma (Fig. 1o). Such protrusions were much more evident in neurons from G592R Kv3.3 knock-in mice, where they could often be seen to contain small membrane-bound organelles (Fig. 1p). Previous work has shown that such protrusions represent an alternative form of clathrin-mediated endocytosis[28].

**Kv3.3 channels bind and activate TBK1.** Because changes in TBK1 (TANK-binding kinase 1) have been linked to some other neurodegenerative conditions[17,29], we tested whether the activity of this enzyme in the cerebellum is altered by the disease-causing G592R Kv3.3 mutation. Western blot analysis of the cerebella of wild type and G592R mutant mice revealed a > twofold increase in levels of phosphorylated TBK1 (pTBK1) in the cerebellum of G592R mutant mice (Fig. 2a, b). In contrast, no change was found in levels of total TBK1 (Fig. 2a, c) or in activated ribosomal S6 kinase (pS6, Fig. 2d, e).

Using CHO cells transfected with wild-type Kv3.3 channels, we found that the channels can be co-immunoprecipitated with

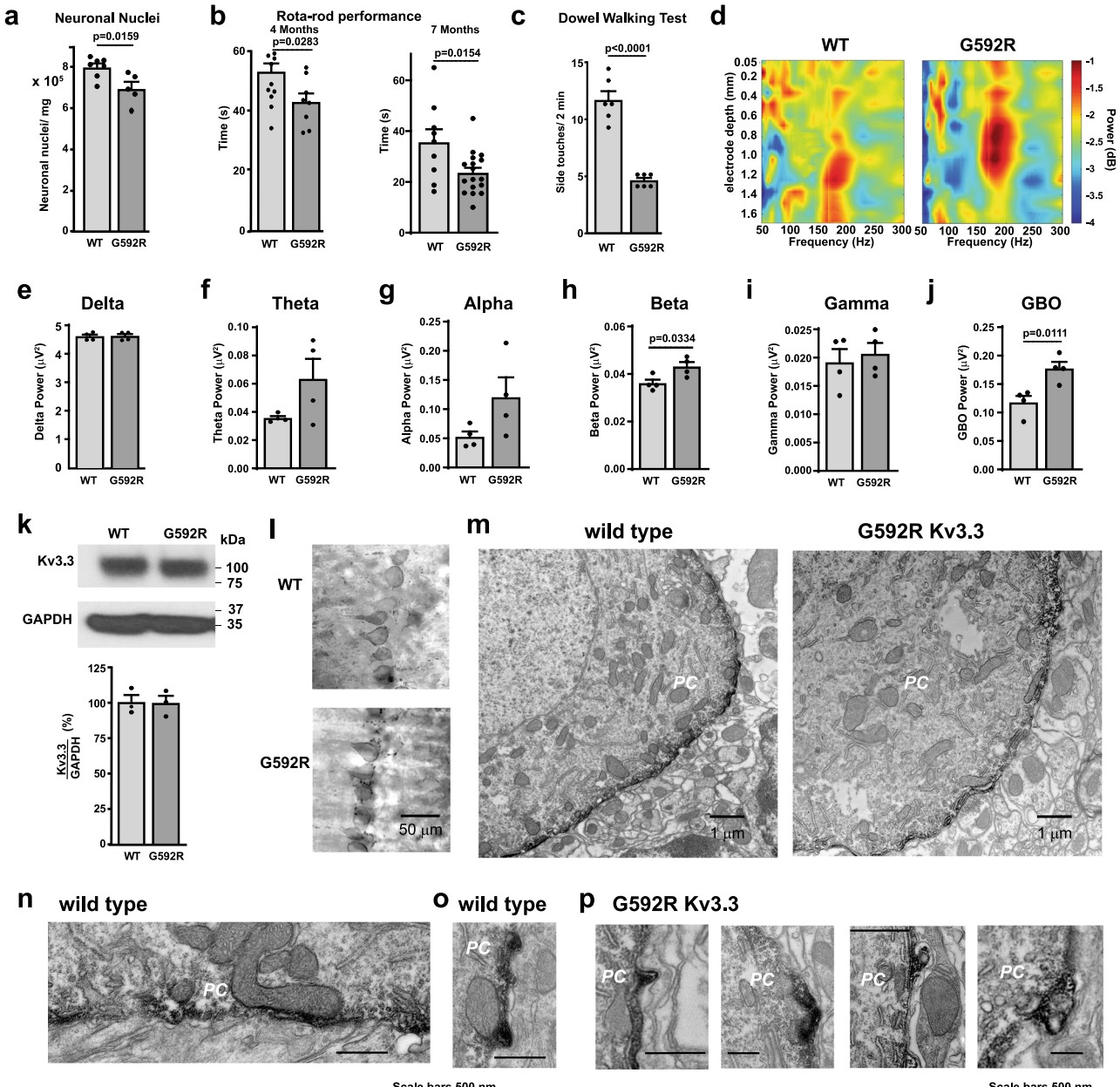

**Fig. 1 Phenotype of G592R Kv3.3 knock-in mice. a** Quantification of numbers of neuronal nuclei per mg of fixed cerebellar tissue ($n = 7$ wild type; $n = 5$ G592R Kv3.3 11-month-old animals. Data are mean ± SEM. Two-tailed unpaired $t$ test). **b** Rota-rod analysis of wild type and mutant mice demonstrating that G592R Kv3.3 mutant animals have significant impairments (wild type at 4 months (females), $n = 13$; G592R mice at 4 months (females), $n = 8$; wild type at 7 months (males), $n = 9$; G592R mice at 7 months (males), $n = 17$). Data are mean ± SEM, two-tailed unpaired $t$ test. **c** Comparison of performance of wild type and mutant mice on the dowel walking test, (wild type, $n = 6$ at 7 months (males); G592R Kv3.3 mice, $n = 6$; at 7 months (males). Data are mean ± SEM, two-tailed unpaired $t$ test. **d** Heat map of power distribution in different frequency bands of local field potentials at different depths in the cerebellar vermis of wild type and mutant mice. Quantification of power in frequency bands; delta (**e**), theta (**f**), alpha (**g**), beta (**h**), gamma (**i**) and GBO (**j**) in wild type and G592R Kv3.3 mice. Data for (**e**–**j**) are mean ± SEM, two-tailed paired $t$ test (wild type, $n = 4$; G592R mutant, $n = 4$). **k** Western blot and group data for Kv3.3 expression in cerebella of wild type and mutant mice. Kv3.3 abundance was normalized to GAPDH (wild type, $n = 3$; G592R mutant, $n = 3$, data are mean ± SEM, two-tailed paired $t$ test). **l** Light level immunostaining of Kv3.3 using diaminobenzidine labeling in cerebellar Purkinje neurons of wild type and mutant mice. **m** Low power EM using diaminobenzidine to localize Kv3.3 in Purkinje cells (PC). **n** Higher power image from a wild type animal showing Kv3.3 immunoreactivity at sites where mitochondria are apposed to the plasma membrane. **o** As n but showing a small Kv3.3-immunoreactive protrusion. **p** Examples of Kv3.3 immunoreactive protrusions, some containing membrane-bound organelles, in Purkinje neurons from G592R Kv3.3 mice. **l**–**p** are representative of 33 and 31 images taken from sections of three wild-type and three mutant mice, respectively. Source data and uncropped Western blots are provided as a Source Data file.

TBK1 (Fig. 2f, g). Moreover, the relative amount of TBK1 that was bound by the G592R Kv3.3 channel was several-fold higher over that bound by wild type Kv3.3 (Fig. 2f, g). By light-level immunomicroscopy, Kv3.3 was found to be closely colocalized with TBK1 in CHO cells (Fig. 2h, i, Pearson's $R$ value 0.98, $p < 0.0001$)[30].

To determine which domains of Kv3.3 are required for its association with TBK1, we generated a series of deletion

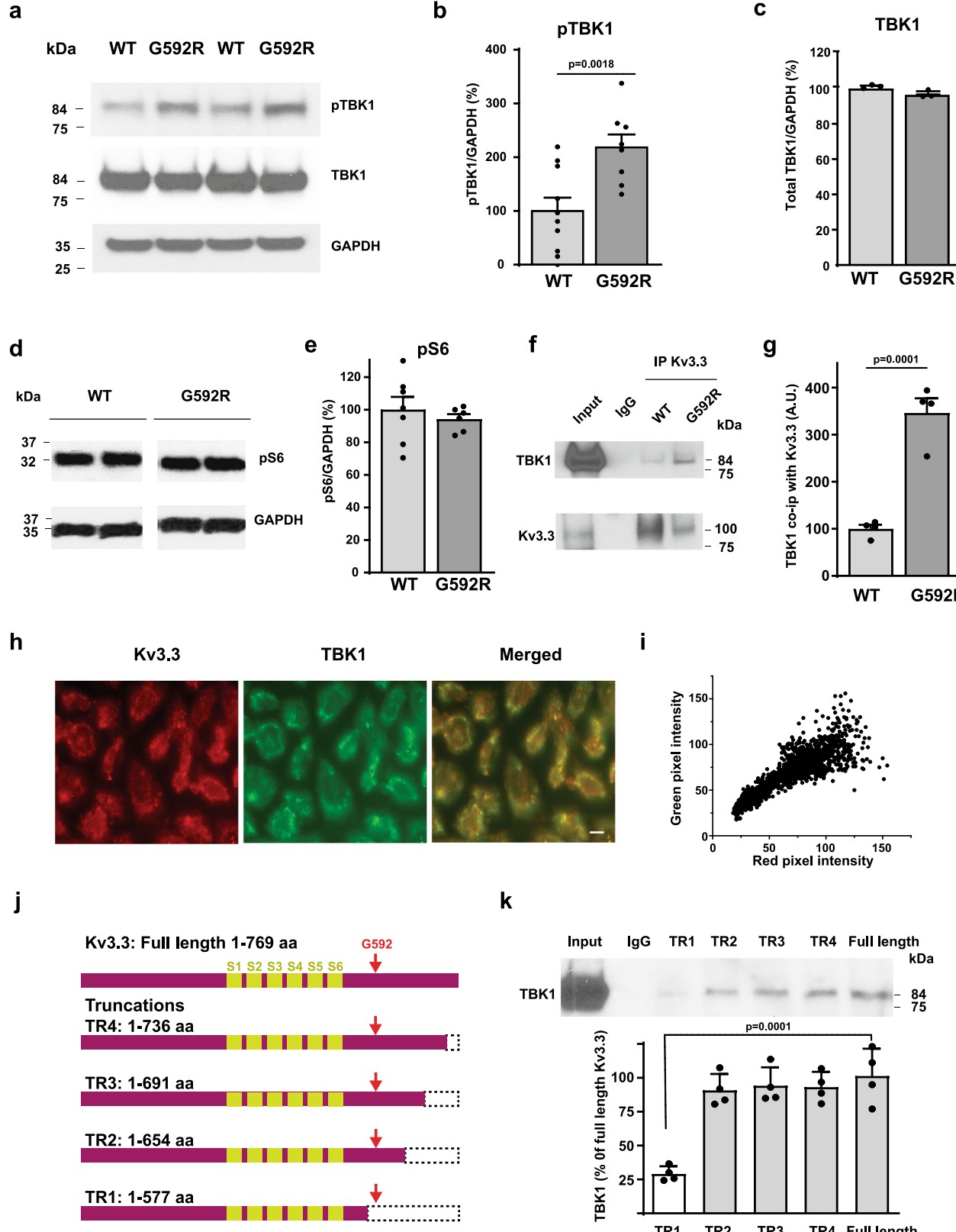

constructs lacking parts of the cytoplasmic C-terminal domain of the channel that were then transfected into CHO cells (Fig. 2j). We found that progressive deletion of up to 115 amino acids from the C-terminus had no effect on the ability of the channel to co-immunoprecipitate TBK1. Further deletion of 77 amino acids, which eliminates the polyproline-rich domain that contains the G592 residue, however, resulted in a major loss of association with the enzyme (Fig. 2k), although this truncation results in a fully functional $K^+$ channel[10].

We next tested whether depolarization of cells expressing Kv3.3 channels alters the activation of TBK1. We compared pTBK1 levels and levels of total TBK1 in cells kept in a medium

**Fig. 2 Kv3.3 channels are linked to TBK1. a** Western blots showing increased pTBK1 in the cerebellum of G592R Kv3.3 mice compared to that in wild type mice, with no change in total TBK1 levels. **b, c** Quantification of pTBK1 and total TBK1 (wild type, $n = 10$; G592R mutant, $n = 8$ independent experiments, two-tailed unpaired $t$ test for p-TBK1; wild type, $n = 3$ independent experiments; G592R mutant, $n = 3$ independent experiments, two-tailed paired $t$ test for total TBK1. Data are mean ± SEM. **d, e** Controls showing no change in levels of active S6 kinase in mutant animals (wild type, $n = 7$ independent experiments; G592R mutant, $n = 6$ independent experiments; data are presented as mean ± SEM, two-tailed unpaired t test). **f** Blots demonstrating that wild type and G592R Kv3.3 channels co-immunoprecipitate (IP) with TBK1. **g** Quantification of relative levels of TBK1 co-immunoprecipitated with wild type and G592R Kv3.3 ($n = 4$ independent experiments; data are mean ± SEM, two-tailed paired $t$ test). **h** Immunostaining demonstrating colocalization of TBK1 (green) with Kv3.3 (red) in CHO cells expressing the channel. Representative images are shown from three independent repeats. Scale bar, 5 μm. **i**, Fluorescence colocalization analysis by pixel intensity. **j** Schematic of truncations (TR) of the C-terminus of Kv3.3. **k** Quantification of the effects of truncations of the Kv3.3 channel C-terminus on the co-immunoprecipitation of TBK1 ($n = 4$ independent experiments; data are mean ± SEM, one-way ANOVA, Tukey's multiple comparisons test). Source data are provided as a Source Data file.

of normal ionic composition with those placed in a depolarizing medium containing elevated $K^+$ ions (90 mM). We found that exposure to the depolarizing medium for only 10 min produced a approximately twofold increase in pTBK1 with no change in total TBK1 levels (Fig. 3a–c). More prolonged incubation in the high $K^+$ medium (5 h) produced no further increase in TBK1 activation and also did not alter total levels of the enzyme (Fig. 3a–c).

To determine whether stimulation of TBK1 is specific to Kv3.3 channels, we repeated this experiment using untransfected cells and cells expressing two other $K^+$ channels: Kv3.1, a channel in the same subfamily as Kv3.3, and KCa1.1 (BK) large-conductance calcium-dependent $K^+$ channels. We also compared cells expressing wild-type Kv3.3 with those expressing the G592R Kv3.3 mutant channels. We found basal pTBK1 levels were in cells expressing wild type Kv3.3, Kv3.1, or BK channels were not different from those in untransfected cells (Fig. 3d). Basal pTBK1 levels in cells expressing G592R Kv3.3 were, however, significantly elevated over those in untransfected or wild-type Kv3.3 cells (Fig. 3d).

Depolarization of untransfected cells produced a small (~20%) increase in pTBK1 (Fig. 3d). Similar small increases in pTBK1 were found in cells expressing Kv3.1 or BK channels, and these increases were not significantly different from those in untransfected cells. In contrast, depolarization of cells containing wild type Kv3.3 resulted in a much larger increase in pTBK1 (Fig. 3d). The greatest activation of TBK1 was found on depolarization of the cells expressing the G592R Kv3.3 channel (Fig. 3d). Thus, the ability of depolarization to produce a two-fold or greater stimulation of TBK1 is specific to Kv3.3 channels and is greatly enhanced by a disease-causing mutation. The effects of depolarization on TBK1 activity did not appear to require ion flux through the channels because they persisted in the presence of 1 mM TEA (tetraethylammonium ions), a potent blocker of all Kv3 family channels (Fig. 3e)[1].

Because TBK1 activation is associated with pathways that result in the turnover of cellular constituents[17], we tested whether depolarization and its associated activation of TBK1 signaling produced any changes in levels of the Kv3.3 channels themselves. Wild type Kv3.3 channels are embedded in a stable cortical actin cytoskeleton that is linked to the channels through the Hax-1 protein[10]. Accordingly, depolarization with elevated $K^+$ for five hours produced no change in levels of Kv3.3 within the cells (Fig. 3f). In contrast, despite binding Hax-1, G592R Kv3.3 channels fail to generate an underlying actin cytoskeleton[10], and treatment of cells expressing G592R Kv3.3 with elevated $K^+$ produced a more than 40% decrease in levels of channel within the cells (Fig. 3g).

**TBK1 activity keeps Hax-1 bound to the Kv3.3 channel**. G592R Kv3.3 channels have been shown to traffic normally to the plasma membrane and to produce $K^+$ currents that are identical to wild type Kv3.3 currents in all parameters except for a slightly slower rate of inactivation during sustained depolarization[10]. Because both wild type and G592R Kv3.3 channels bind Hax-1, we tested whether TBK1 activity alters the ability of these channels to interact with Hax-1. We first tested co-immunoprecipitation of Hax-1 with the channels in the presence and absence of MRT67307, an inhibitor of TBK1[31,32]. Preincubation of cells for 30 min with 10 μM MRT67307 was found to greatly reduce levels of Hax-1 coimmunoprecipitated with either wild type or G592R Kv3.3 channels (Fig. 4a).

Because the binding of Hax-1 to Kv3.3 prevents rapid inactivation of the channels[10], we tested the effects of the TBK1 inhibitor on the characteristics of Kv3.3 currents using the whole-cell patch-clamp technique. Under control conditions, wild type Kv3.3 channels inactivate only slowly and partially during a depolarization to positive potentials lasting 600 msec (Fig. 4b, *left panel*). Treatment with 10 μM MRT67307 for 30 min resulted in rapid and complete inactivation of both wild type and G592R mutant channels (Fig. 4b, *right panels*). Consistent with the finding that a major component of inactivation of Kv3.3 channels represents N-type inactivation, mediated by entry of the cytoplasmic N-terminal ball into the conduction pathway[10,33], the effects of MRT67307 measured 100 ms after the onset of depolarization were absent in Kv3.3 channels truncated at the N-terminus for all concentration of the inhibitor up to 20 μM (ΔN1-78 Kv3.3, Fig. 4c *left panel*). With more prolonged depolarization (600 msec), however, degree of inactivation of the N-truncated mutant was not statistically different from that of wild type or G592R Kv3.3 (Fig. 4c, *right panel*), suggesting that TBK1/Hax-1 also regulates other aspects of channel inactivation.

These findings suggest that a basal level of TBK1 activity is required to keep Hax-1 bound to the channel and to prevent channel inactivation. As a further test of this hypothesis, we overexpressed TBK1 in cells expressing the channels. We found that the rate of inactivation for both wild type and G592R Kv3.3 channels was substantially slowed when co-expressed with TBK1 (Fig. 4d–f). These findings are consistent with the hypothesis that TBK1-dependent association of Kv3.3 with Hax-1 regulates its rate of inactivation (Fig. 4g) and the finding that inactivation of G592R Kv3.3 channels is slower than that of wild type channels (Fig. 4h).

**Mutant Kv3.3 channel traffics Hax-1 into the late endosome/lysosomal pathway**. Because TBK1 plays a role in the envelopment of both invading and endogenous cellular constituents, we compared Hax-1 localization in untransfected CHO cells with that in cells expressing wild type Kv3.3, and those expressing the G592R mutant channels, which have elevated TBK1 activity (Fig. 5). At the light microscopic level, we found that, Hax-1 immunoreactivity in untransfected CHO cells is localized to a single discrete location in each cell (Fig. 5a). In contrast, as

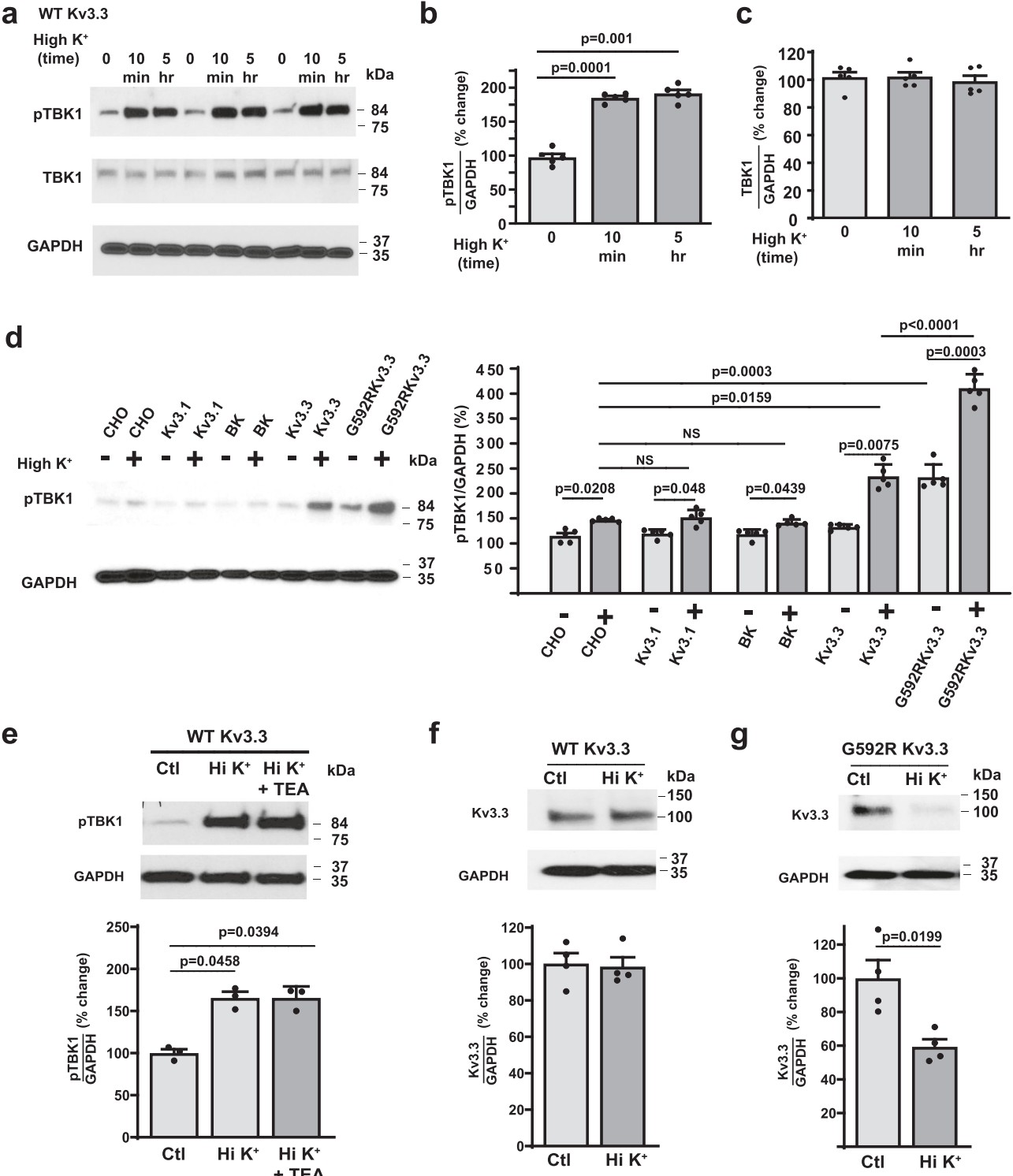

reported previously[10], Hax-1 is recruited uniformly under the plasma membrane in cells expressing wild type Kv3.3 (Fig. 5b). In addition, the overall level of Hax-1 immunoreactivity appeared higher in wild-type Kv3.3-expressing cells than in untransfected cells, a finding confirmed by quantification of pixel intensity in fields of Kv3.3-expressing cells (Fig. 5g). In contrast, Hax-1 immunostaining in cells expressing G592R Kv3.3 was reduced compared to that in cells expressing the wild-type channel (Fig. 5g). In many of the cells expressing G592R Kv3.3, Hax-1 had a similar membrane localization to that in wild-type Kv3.3 cells

(Fig. 5c, *right panel, green asterisks*). A proportion of the cells expressing the mutant channels, however, exhibited a very dense pattern of staining. These cells were also conspicuous by being surrounded by numerous small deposits of Hax-1-stained extracellular structures (Fig. 5c, *left panel, red arrows*). The densely stained cells represented ~8–9% of the population of cells (Fig. 5h).

Immuno-EM revealed that, in the densely stained G592R Kv3.3 expressing cells, Hax-1 is present in undigested inclusions within large late endosomal/lysosomal structures (Fig. 5f). In

**Fig. 3 Depolarization of cells expressing Kv3.3 channels activates TBK1. a** Immunoblots of pTBK1 in cells expressing wild type Kv3.3 cultured with normal extracellular medium for 10 min or for 5 h. All blots were probed with antibody against pTBK1(*top*) and then stripped and reprobed for total TBK1 (*center*) and then again for GAPDH (*bottom*). Bar graphs showing the quantification of pTBK1 levels (**b**) and total TBK1 (**c**) among the three groups. In all blots, line scanning densitometry was performed for pTBK1, TBK1 and GAPDH probed blots and then pixel quantification for pTBK1 and TBK1 was normalized to the sample loading control GAPDH. Values are mean ± SEM ($n = 5$ independent experiments, ANOVA, Dunnett's multiple comparisons test). **d** Representative Western blots (*left*) with quantification (*right*) showing the changes in pTBK1 levels in untransfected Chinese Hamster Ovary (CHO cells) or cells expressing Kv3.1, BK channels, wild type Kv3.3 or G592R Kv3.3 after depolarization (10 min) with high extracellular $K^+$ (90 mM, Hi $K^+$). ($n = 5$ for each group; data are presented as mean ± SEM, ANOVA, Tukey's multiple comparison test). **e** The presence of a Kv3.3 channel blocker (1 mM tetraethylammonium (TEA)) did not prevent the increase in pTBK1 after high $K^+$ depolarization for 5 h in cells expressing wild type (WT) Kv3.3 cells ($n = 3$ independent experiments; data are presented as mean ± SEM, ANOVA, Tukey's multiple comparison test). Prolonged depolarization reduces levels of G592R Kv3.3 but not wild type Kv3.3 channels. Representative Western blots with quantification showing levels of Kv3.3 in CHO cells expressing wild type Kv3.3 (**f**) and G592R Kv3.3 channels (**g**) after depolarization (5 hrs) with high extracellular $K^+$ (90 mM) ($n = 4$ independent experiments; data are presented as mean ± SEM, two-tailed paired $t$ test). Source data and uncropped Western blots are provided as a Source Data file.

untransfected cells and those expressing wild type Kv3.3, Hax-1 immunolabeling was confined to a discrete region at the plasma membrane or uniformly distributed at the membrane, respectively (Fig. 5d,e). In G592R Kv3.3 cells, Hax-1 immunoreactivity was also present at the plasma membrane but also found in structures composed of Hax-1 containing organelles surrounded by a second membrane (Fig. 5f). Such organelles could also be found at the surface of cells, consistent with their being extruded from the cells (Fig. 5f, *right* panel).

To confirm the interpretation of the EM immunolabeling profiles, we measured levels of CD63, in the cell cultures subjected to prolonged depolarization (5 h, 90 mM $K^+$). CD63 is a tetraspanin molecular marker for multivesicular bodies/late endosomes that can be trafficked to the lysosomes or to the plasma membrane to release intraluminal vesicles as exosomes. Levels of CD63 in CHO cells expressing wild type Kv3.3 were not different from those in untransfected cells but were elevated over 2.5-fold in cells expressing G592R Kv3.3 (Fig. 5i). This increase in CD63 was reversed by pretreatment of the cells with the TBK1 inhibitor 10 μM MRT67307 for 30 min, supporting a causal role for enhanced TBK1 activity in the formation of multivesicular bodies.

By visual inspection, the viability of G592R Kv3.3-expressing cells with numerous Hax-1-containing inclusions appeared compromised. We, therefore, measured their rate of cell death using propidium iodide (PI) staining and compared this with that of untransfected CHO cells and those expressing wild type Kv3.3. No difference in levels of PI was found between untransfected cells and those with the wild-type channel, but PI staining was markedly elevated in the G592R Kv3.3 cells (Fig. 6a). This was associated with an increase in levels of the executioner caspase-7 (Fig. 6b,c). As with the multivesicular body marker CD63, the increase in PI signal and the activation of caspase 7 were reduced by inhibition of TBK1 (Fig. 6a,b,c). Because Hax-1 is a cell survival protein and levels of Hax-1 are reduced in G592R Kv3.3 cells compared to cells with wild-type Kv3.3 channels (Fig. 5g), we tested whether overexpression of Hax-1 in G592R Kv3.3 cells could rescue the effects of this mutation. Levels of cleaved caspase-7 were found to be very markedly reduced in G592R Kv3.3 cells transfected with Hax-1 over those with a control vector (Fig. 6d, e). These findings support the hypothesis that overactivation of TBK1 promotes cell death by increasing the targeting of Hax-1 to the late endosomal/lysosomal pathway.

**Mutant Kv3.3 reduces Hax-1 levels in cerebellar neurons in vivo.** In order to determine if these findings in cell culture are relevant to neurons in vivo, we compared levels of Hax-1 in the cerebellum of wild-type mice and those bearing the G592R Kv3.3 mutation. We found that levels of this cell survival protein were reduced close to 50% by the channel mutation (Fig. 7a). In

addition, levels of activated caspase 7 were increased by the mutation (Fig. 7b). As was also found in the cell lines, levels of the multivesicular body marker CD63 were increased over three-fold in the cerebellum of animals expressing the mutant channel,over those in wild-type mice (Fig. 7c). We also tested for changes in two other markers of intracellular trafficking linked to lysosomes. Levels of LAMP2 (lysosomal associated membrane protein-2), a marker of lysosomal content were increased by the G592R Kv3.3 (Fig. 7d). In contrast, no change was found in levels of LC3BII (microtubule-associated protein 1 A/1B-light chain 3), a marker for autophagy (Fig. 7e).

Consistent with the finding of increased numbers of multivesicular bodies in cell lines expressing G592R Kv3.3 and with elevated levels of CD63 in both the cell lines and the cerebellum of animals expressing this mutation, electron micrographs of the somata of Purkinje cells revealed an approximately twofold increase in the numbers of multivesicular bodies in the cytoplasm of the mutant mice compared to those in wild type (Fig. 7f, g). The numbers and cytoplasmic density of lysosomes were not different in Purkinje cells from mutant and wild-type animals (Fig. 7h, i). In contrast to the wild type, however, over 90% of lysosomes in cells from G592R mice contained dense inclusions (Fig. 7f, j), consistent with the finding that lysosomes lacking such inclusions (light lysosomes) was reduced by ~80% in the mutant cells (Fig. 7j). We therefore next used immunoEM to determine whether, as in the transfected cell lines, the distribution of Hax-1 in Purkinje neurons was altered by the G592R Kv3.3 mutation. In neurons from wild-type animals, immunostaining could be detected at the plasma membrane (Fig. 8a, b) and, like Kv3.3, higher levels of immunoreactivity were detected at sites where mitochondria come in close apposition to the plasma membrane (Fig. 8c, d). Consistent with findings in other cell types[34], Hax-1 could also be found associated with endoplasmic reticulum (Fig. 8e). As was also found for Kv3.3, small Hax-1-immunoreactive protrusions could occasionally be detected jutting out of the Purkinje cell somata of neurons from wild-type animals (Fig. 8f). In sections of Purkinje cells from G592R Kv3.3 knock-in mice, the same general pattern of Hax-1 immunolocalization was found at the plasma membrane (Fig. 8g) and at sites where this membrane contacts mitochondria (Fig. 8h), as well as on the endoplasmic reticulum (Fig. 8i, j). In contrast to wild-type animals, however, intracellular organelles could be detected in which membrane-associated Hax-1 immunoreactivity was enveloped by a second membrane (Fig. 8k, l). These findings are consistent with enhanced trafficking of Hax-1 into multivesicular bodies/late endosomes in Purkinje cells of G592R Kv3.3 knock-in mice.

**The Kv3.3 G592R channel stimulates the production of exosomes.** We have shown that, in Purkinje neurons, the G592R

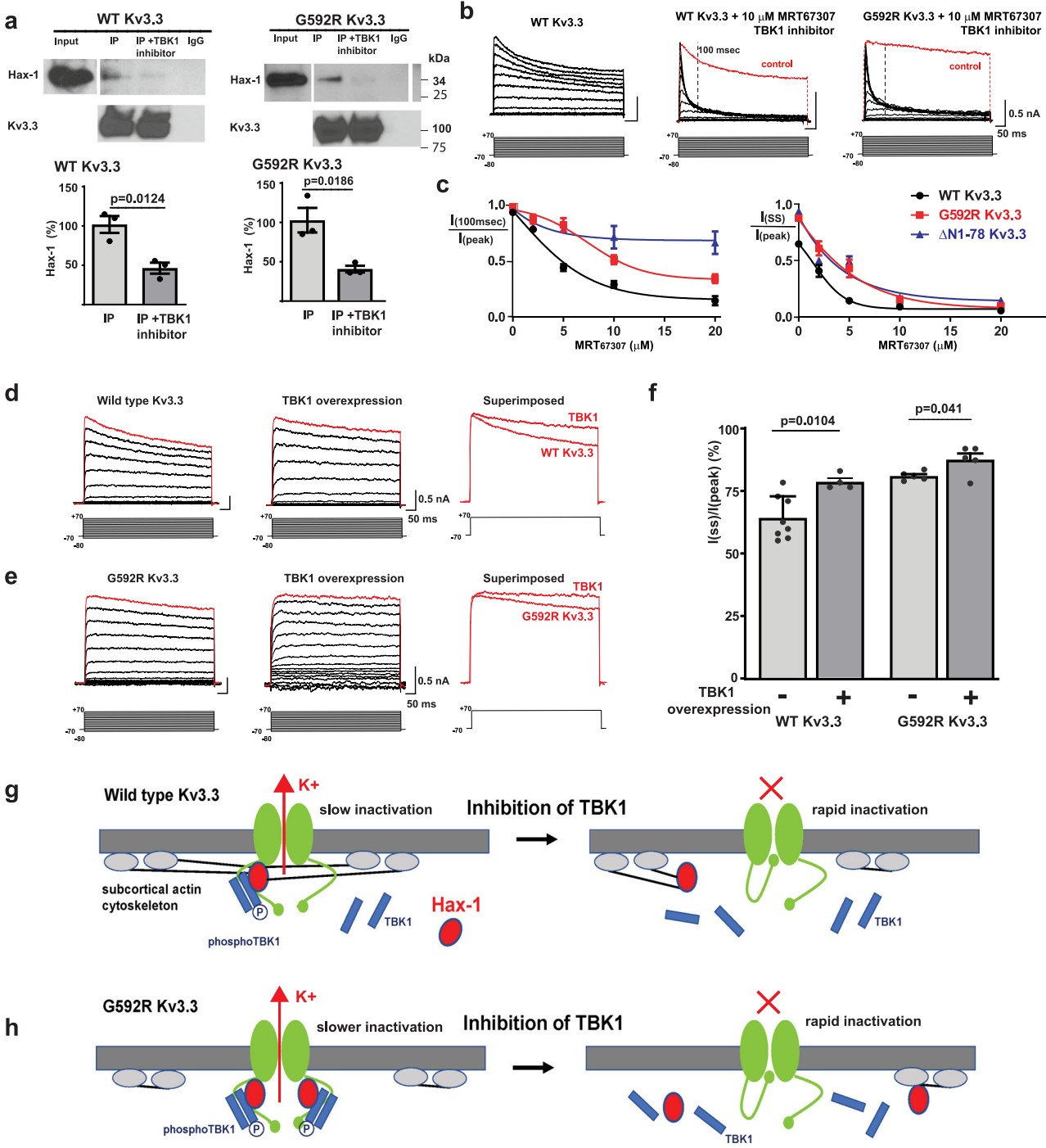

Kv3.3 mutation causes endolysosomal dysfunction and increased levels of CD63, a marker of multivesicular bodies that can be trafficked to lysosomes or fuse with the plasma membrane, releasing exosomes into the extracellular space. To test whether levels of exosomes are altered in Kv3.3 G592R mutant mice, we isolated EVs from cerebella of G592R Kv3.3 and wild-type mice and fractioned EVs using an iodixanol-based step gradient[35]. This separates low-density fractions (LDFs), which contain mainly microvesicles identified by the marker Annexin A2[36], from intermediate density fractions (IDFs), which contain exosomes (Fig. 9).

Under transmission electron microscopy, classical biconcave and cup-shaped EVs were seen in both fractions, with no contaminating intracellular organelles or debris (Fig. 9a). Western Blot analyses of equal amounts of EV lysates revealed that CD63 (a marker of exosomes) and Annexin A2 (found exclusively in microvesicles) were enriched in cerebellar EVs when compared to the cerebellar homogenate and were enriched in IDFs and LDFs, respectively. As a control, Flotilin-1, a protein abundant in both exosomes and microvesicles[37], was equally distributed (Fig. 8b, *left* panel). Calbindin-D28k, a marker of Purkinje cells in the cerebellum, was detected in both microvesicles and exosomes at approximatively the same level (Fig. 9b, *left* panel). We further corroborated the quality of our isolation method demonstrating; i) the absence of intracellular organelle contaminants, including nuclei, Golgi and endoplasmic reticulum (with Lamin-A/C,

**Fig. 4 TBK1 activity regulates Kv3.3 channel inactivation. a** Western blots (top) demonstrating that co-immunoprecipitation of Hax-1 with either wild type or G592R Kv3.3 is prevented by a TBK1 inhibitor (MRT67307, 10 μM); Bottom panels show quantification of the relative levels of Hax-1 co-immunoprecipitated with Kv3.3 with and without inhibitor (values are mean ± SEM; $n = 3$ independent experiments, two-tailed unpaired $t$ test). **b** Representative voltage clamp traces showing that TBK1 inhibition produces rapid inactivation of wild type and G592R Kv3.3 currents. A dashed line indicates 100 msec after the onset of depolarization. **c** Concentration-response plots for effects of TBK1 inhibitor on inactivation. Left plot quantifies inactivation at +70 mV, 100 ms after onset of depolarization ($I_{(100\,ms)}/I_{(peak)}$), while right panel shows amounts of inactivation at the end of the 600 ms depolarizing pulses ($I_{(SS)}/I_{(peak)}$). Also shown are data for the ΔN1-78 Kv3.3 mutant ($n = 4$ independent experiments, data are mean ± SEM). **d, e** Representative voltage-clamp traces showing that overexpression of TBK1 slows inactivation of wild type and G592R Kv3.3 currents. Traces at +70 mV are shown in red and superimposed at right for comparison. **f** Quantification of effect of TBK1 overexpression on inactivation measured 600 ms after depolarization to +70 mV for wild type and mutant channels (values are mean ± SEM; TBK1 overexpression in wild type, $n = 4$; vector alone in wild type, $n = 8$; TBK1 overexpression in G592RKv3.3 cells, $n = 5$; vector alone in G592R Kv3.3 cells, $n = 5$, two-tailed paired $t$ test). **g** and **h**, Potential interactions between Kv3.3 and TBK1 activity. Basal TBK1 activity is required for Hax-1 binding to the Kv3.3 C-terminus, which leads to formation of a channel-associated actin cytoskeleton (*g left*). Inhibition of TBK1 (*g right*) causes dissociation of Hax-1, allowing the Kv3.3 cytoplasmic N-terminus to occlude the pore (inactivation). G592R Kv3.3 channels bind more total TBK1 and pTBK1 than wild-type channels (*h left*) and have slower inactivation. As with wild-type channels, inhibition of TBK1 (*h right*) causes dissociation of Hax-1 and rapid inactivation. Source data and uncropped Western blots are provided as a Source Data file.

GM-130, and Sec 61B as markers for each organelle), ii) lack of cytosolic components such as β-actin[36], and iii) effective elimination of lipoproteins, such as VLDLs and LDLs (Apo-B used as a marker), common contaminants of EV preparations (Fig. 9b, *right* panel).

Microvesicles tend to be larger than exosomes and by visual inspection we observed some vesicles with a large diameter in LDFs (Fig. 9a). Nanotrack analysis (NTA) for the quantification of EV dimensions revealed that the hydrodynamic diameter of LDF EVs peaked at 150 nm, while the corresponding value for IDF EVs was 100 nm (Fig. 9c), demonstrating a higher percentage of small EVs (<100 nm) and a lower percentage of larger EVs (>150 nm) in IDFs compared to LDFs (Fig. 9d). These data confirmed the presence of larger vesicles in LDFs vs IDFs, consistent with the microvesicle vs exosome nature of the vesicles enriched in these fractions.

We next used Western blotting to compare levels of EV markers in cerebella of Kv3.3 G592R and wild type mice. The levels of three different markers for exosomes (HSC70, Alix, and CD63) were higher in the IDFs of Kv3.3 G592R mice (Fig. 9e, f), while no significant difference was detected for Annexin A2 between the genotypes (Fig. 9e, g), indicating that the release of exosomes, but not microvesicles, is specifically increased in cerebella of the mutant mice. Calbindin-D28k, equally distributed between LDFs and IDFs in wild type mice (Fig. 9b), showed higher levels in the exosomal fractions of G592R Kv3.3 mutant mice (Fig. 9e, h). This further indicates that Purkinje cells in mutant mice release more exosomes than the controls, while microvesicles are relatively unaltered. No immunoreactivity for Kv3.3, TBK1 or Hax1 could be detected in cerebellar EVs (Supplementary Fig. S2).

We also performed global quantification of EVs using both total protein amount estimation and total particle number measure calculated by NTA, as suggested by the MISEV2018 guidelines[38] (Fig. 9i, j). Consistent with our Western blot analyses, these showed no difference in the LDF EV levels between the genotypes. Both protein and particle levels were, however, higher in the IDF exosome-enriched fractions of the mutant mice compared to controls. The findings indicate that the G592R Kv3.3 mutation increases exosome secretion in the cerebellum in vivo, while microvesicle levels are unaffected.

## Discussion

TBK1 is a serine/threonine kinase, that regulates multiple cellular processes and that, when dysfunctional, can trigger neurodegeneration[17,39,40]. We have found that the neuronal Kv3.3 voltage-dependent K$^+$ channel is intimately linked to the activity of TBK1. Both short-term and long-term depolarization of cells expressing Kv3.3 produces activation of this enzyme. A likely mechanism for this activation is that, on depolarization, the channel complex provides a platform for binding two or more TBK1 monomers, allowing activation by transphosphorylation[32]. While we have defined regions of the cytoplasmic C-terminus of Kv3.3 required for its interaction with TBK1, we cannot be certain whether the interaction is direct or requires additional channel-binding partners. A human mutation in the Kv3.3 channel that results in late-onset cerebellar degeneration (SCA13) further enhances the activation of TBK1 both in cell lines and in the cerebellum of mice expressing this mutation. Mutations in TBK1 itself have previously been associated with another neurodegenerative disease, familial ALS, as well as with fronto-temporal dementia[29], although many of these mutations lead to deficits in TBK1 activity rather than overactivity as with the G592R Kv3.3 mutation.

Figure 10, together with Fig. 4g,h, provides a tentative outline of the interactions we have described. TBK1 activity is required for binding of Kv3.3 channels to Hax-1, an anti-apoptotic protein that is required for survival of cerebellar neurons[11,12] (Fig. 4g, *left*). This cell survival protein also regulates the formation of new actin filaments by the Arp2/3 complex[13], and the interaction of wild type Kv3.3 channels with Hax-1 leads to the formation of a stable channel-associated subcortical actin cytoskeleton[10]. The binding of Hax-1 to the C-terminus of Kv3.3 channels also prevents rapid inactivation of the channel during depolarization. We have now found that the binding of Hax-1 requires a basal level of TBK1 activity and that inhibition of TBK1 produces rapid channel inactivation (Fig. 4g, *right*). This rapid inactivation mimics that produced by suppression of Hax-1 expression[10]. G592R Kv3.3 channels inactivate more slowly than wild type Kv3.3 channels, consistent with the finding that more TBK1 activity is associated with this mutant channel (Figs. 2h, 3d and 4h) and that overexpression of TBK1 further slows the inactivation of both wild type and mutant channels (Fig. 4e, f). Thus, TBK1 is an endogenous regulator of the Kv3.3/Hax-1 interaction.

Prolonged depolarization of cell lines expressing Kv3.3 results in a loss of G592R Kv3.3 channels but not of wild type channels. This spinocerebellar ataxia mutant channel also promotes the loss of Hax-1, as also occurs in Friedreich's ataxia, another disorder that results in the degeneration of neurons in motor systems including the cerebellum[41]. One difference between the mutant and wild type channels is that the latter are embedded in a stable subcortical actin cytoskeleton, while the mutant Kv3.3/Hax-1 complexes are not bound to the actin cytoskeleton (Fig. 4g, h)[10]. Using both light level and EM immunostaining we found that the trafficking of Hax-1 in cells expressing G592R Kv3.3 is abnormal.

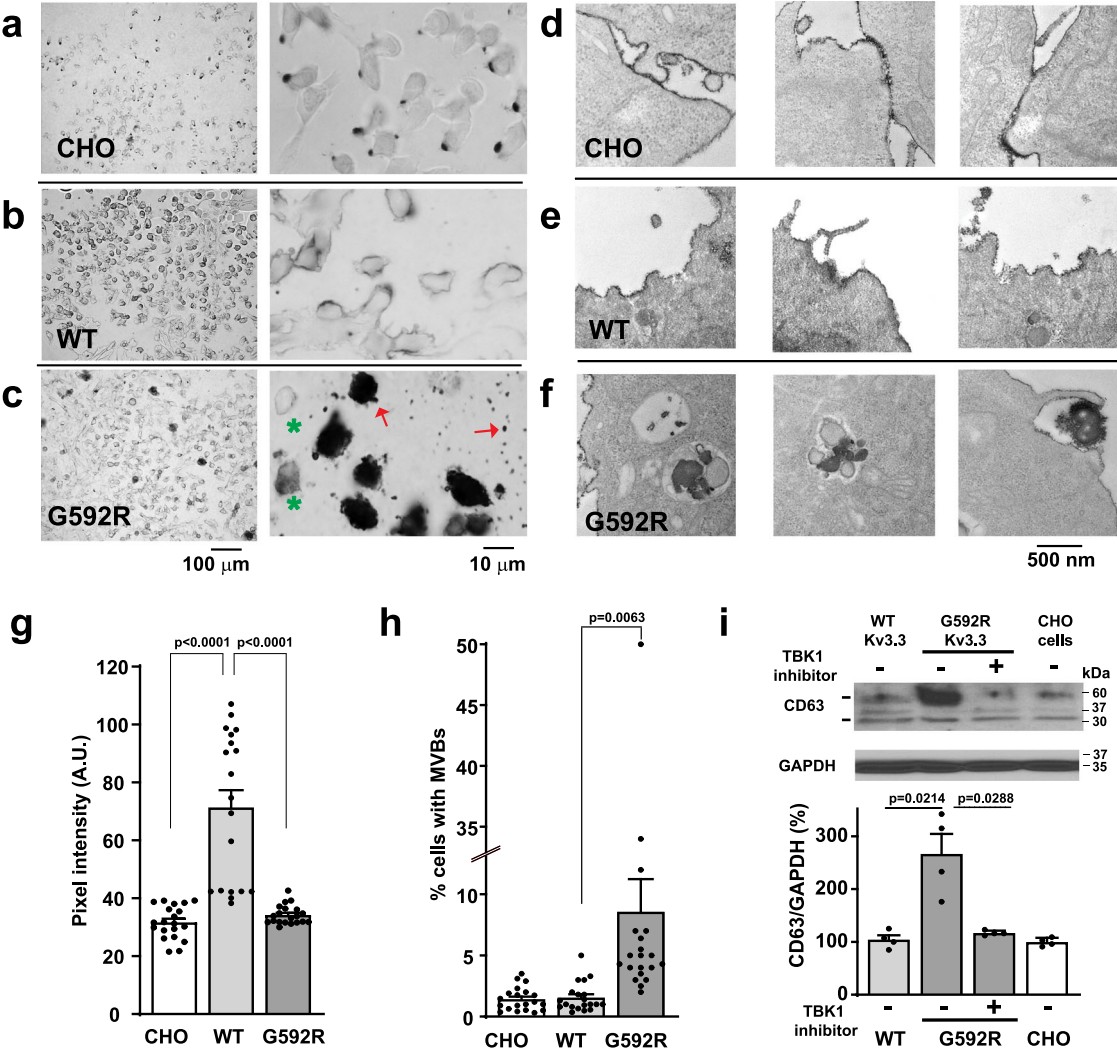

**Fig. 5 The G592R Kv3.3 channel causes trafficking of Hax-1 into inclusions within lysosomes.** Light level immunolocalization of Hax-1 in untransfected CHO cells (**a**) and cells expressing wild type (**b**) or G592R Kv3.3 (**c**). *Left* panels show low power views and *right* panels show Hax-1 immunostaining at higher magnification. Scale bars under **c** apply to all pairs of panels in (**a–c**). In cells expressing G592R Kv3.3, the peripheral staining for Hax-1 resembles that of cells expressing the wild-type channel (green asterisks in right panel of **e**). With the mutant channel, however, a high proportion of cells develop dense Hax-1 inclusions (multivesicular bodies), associated with extracellular deposits (red arrows in right panel of **e**). Images are representative of 20 independent fields of cells. Electron immunolocalization of Hax-1 in untransfected CHO cells (**d**) and those expressing wild type (**e**) or G592R Kv3.3 (**f**). Hax-1 staining is found uniformly under the plasma membrane in cells expressing wild type Kv3.3. In G592R Kv3.3 cells, Hax-1 is localized not only to the plasma membrane but in large late endosomal/lysosomal structures. Panels are representative of 23, 20, and 87 images of untransfected CHO, wild-type Kv3.3 and G592R Kv3.3 cells, respectively. **g** Quantification of overall intensity of Hax-1 immunostaining in plates of untransfected CHO cells and cells expressing wild type or G592R Kv3.3 ($n = 20$ fields. Data are presented as mean ± SEM, one way ANOVA, Tukey's multiple comparison test). **h** Quantification of the percentages of cells with dense Hax-1-containing inclusions in the three cell types ($n = 20$ fields of cells in each condition. Data are presented as mean ± SEM. One-way ANOVA, Tukey's multiple comparisons test. **i** Immunoblots with quantification of the increase in levels of multivesicular body marker CD63 in cells expressing mutant Kv3.3. This increase was reversed by pretreatment of the cells with the TBK1 inhibitor 10 μM MRT67307 ($n = 4$ independent experiments; data are presented as mean ± SEM. Two-tailed paired $t$ test). Source data and uncropped Western blots are provided as a Source Data file.

In cells expressing wild type Kv3.3, Hax-1 is recruited to the plasma membrane with the channel. In contrast, in cells expressing the mutant channel, a significant amount of Hax-1 becomes delocalized from the plasma membrane and accumulates in dense late endosomal/lysosomal structures.

In both transfected cells and in the cerebellum of G592R mice, the increased activation of TBK1 is associated with a marked increase in levels of CD63, a marker for multivesicular bodies, which then target their cargo for lysosomal degradation or for expulsion from the cells as exosomes (Fig. 8e). Higher levels of CD63 expression have also been found in the brain of Down

syndrome patients and a mouse model of the disease, and as in the present study, are associated with a higher amount of multivesicular bodies and with higher levels of exosomes in the extracellular space[42,43]. Interestingly, we did not detect Kv3.3 or Hax-1 in the exosomes released into the extracellular space in the Kv3.3 mutant animals, although we cannot eliminate the possibility that they are present in these organelles at lower levels or degraded before release. This suggests that multivesicular bodies containing these proteins are selectively targeted to lysosomes. Exosomal and endosomal pathways are interconnected and modulate each other[42–46]. Thus, the higher release of exosomes in G592R animals

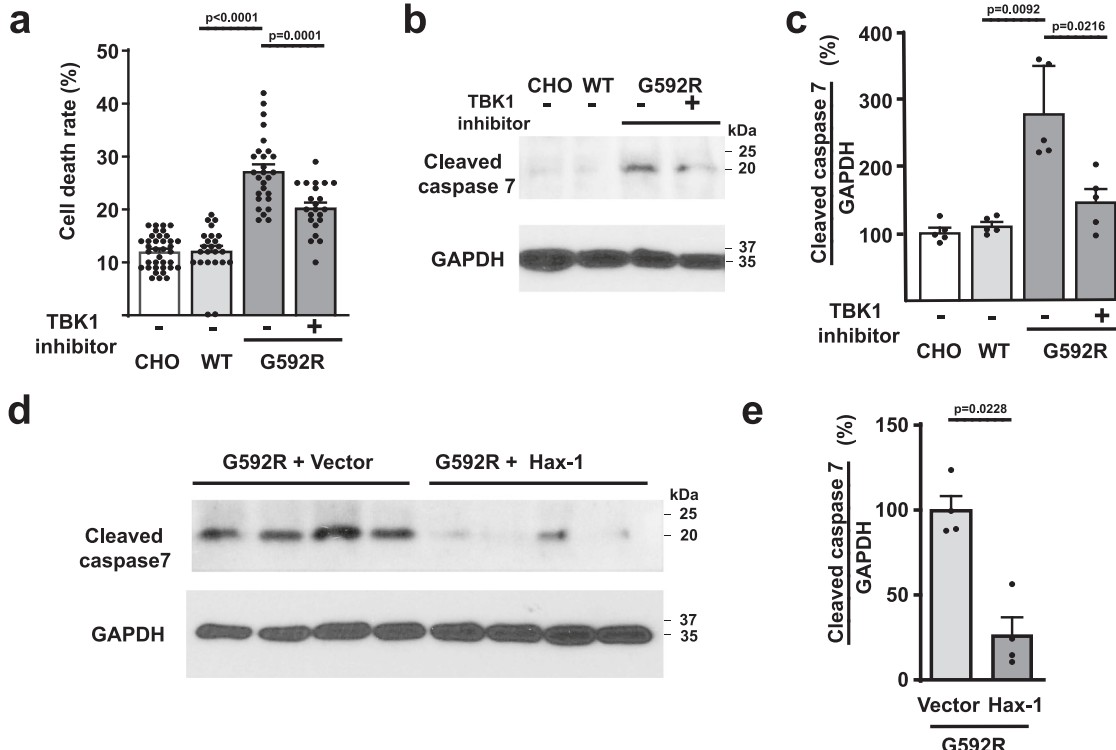

**Fig. 6 Cell death is increased by G592R Kv3.3. a** Quantification of propidium iodide staining in untransfected CHO cells and cells expressing wild type or G592R Kv3.3 channels ($n = 3$ dishes; data are presented as mean ± SEM; two-tailed unpaired $t$ test). **b** Western blots showing levels of cleaved caspase 7 in the three groups of cells and in G592R Kv3.3 cells pretreated with the TBK1 inhibitor 10 μM MRT67307. **c** Group data for **b** ($n = 5$ independent experiments; data are presented as mean ± SEM; two-tailed paired $t$ test). **d** Western blots showing the effect of overexpression of Hax-1 in G592R Kv3.3 cells on levels of cleaved caspase-7. **e** Group data for $d$ ($n = 4$ independent experiments; data are presented as mean ± SEM, two-tailed paired $t$ test). Source data and uncropped Western blots are provided as a Source Data file.

may represent a TBK1-regulated homeostatic mechanism to alleviate intracellular endosomal abnormalities, as was previously suggested for other neurodegenerative conditions[42,43,46].

While it is possible that the G592R mutation results in misfolding of the Kv3.3 channel and that this directly alters the trafficking of mutant protein, current evidence suggests that the mutation simply enhances trafficking through a physiological pathway that controls the normal turnover of Kv3.3 channels. Evidence supporting the latter view is that the wild type Kv3.3 channel, but not the closely related Kv3.1 or the BK channel, also selectively activates the TBK1 pathway. Moreover, the G592R Kv3.3 mutant channel expresses normal Kv3.3 currents and is trafficked normally to the plasma membrane (Figs. 1 and 4)[10]. As far as we are aware, there is no evidence that TBK1 itself recognizes mutated or misfolded proteins. Thus, it is unlikely that pathological misfolding contributes to the Kv3.3-dependent activation of the TBK1 pathway.

In summary, our findings indicate that TBK1 regulates neuronal excitability by determining whether Kv3.3 channels remain open during prolonged depolarizations. Because, in turn, Kv3.3 stimulates TBK1 enzyme activity, activation of these channels during neuronal firing may directly control the turnover of neuronal components, such as their associated cell survival protein Hax-1, potentially leading to the eventual demise of neurons expressing channel mutations that overstimulate this pathway.

## Methods

**Cell culture**. CHO cells were grown in Iscove's modified Dulbecco's medium (Invitrogen) supplemented with 10% fetal bovine serum (heat-inactivated), 100 units/ml penicillin/streptomycin, 5% HT supplement (Invitrogen) in a 5% $CO_2$ incubator at 37 °C. For experiments testing the effects of depolarization, cells were transferred to either a control physiological medium consisted ((in mM) 140 NaCl,

5.4 KCl, 1.3 $CaCl_2$, 25 HEPES, 33 glucose, pH 7.4, with NaOH) or a high $K^+$ medium ((in mM) 55.4 NaCl, 90 KCl, 1.3 $CaCl_2$, 25 HEPES, 33 glucose, pH 7.4, with NaOH), and incubated for times indicated in the text.

**Generation of G592R Kv3.3 knock-in mice by Crispr/Cas 9 genome editing**. To generate the G592R Kv3.3 point mutant mouse strain, we used the CRISPR/ Cas9 system[47]. In brief, a gRNA close to the genomic location of the desired mutation was designed to maximize cutting efficiency and minimize off-target effects using the CRISPOR algorithm. Cas9 mRNA, a single guide RNA (gRNA), and a repair template including the point mutation were in vitro transcribed. Superovulated female C57BL/6J mice were mated to stud C57BL/6J males, and fertilized embryos collected from the oviducts. The Cas9 mRNA, gRNA, and repair template were injected into the cytoplasm of ~175 fertilized eggs and then transferred into uterus of pseudopregnant female mice (gRNA sequence: CACCCC CACCACGGCAGCGGGTTTTAGAGCTAGAAATAGCAAGTTAAAATAAG GCTAGTCCGTTATCAACTTGAAAAAGTGGCACCGAGTCGGTGCTTTTTT; Repair template: GCAGCCTGGCTCACCCAACTACTGCAA GCCTGACCCCCGCCTCCACCCCCACCACACCCCCACCACGGCAGCC GTGGCATAAGCCCACCGCCGCCCATCACCCCTCCTTCCATGGGGGTGAA TGT). F0 founder mice were screened for the insertion of the point mutation and the absence of additional mutations in the nearby regions to the target sequence for the selected gRNA to ensure adequate DNA repair of the specific genomic locus. Out of nine F0 founders, two containing the desired point mutation and perfect repair of the locus were identified using standard PCR and sequencing methodologies. One of these positive F0 founders bearing a heterozygous mutation was bred to C57BL/6J wild-type mice for three generations to obtain heterozygous mice and breed out potential off target mutations. Heterozygous mice were then bred to each other to obtain littermate mutant and wild-type mice for experimentation.

**Genotyping analysis**. DNA was extracted with DNeasy Blood & TissueKit (Qiagen). PCR was performed using Platinum Taq DNA Polimerase High Fidelity (Qiagen) followed by a PCR purification (Qiagen). DNA concentration was measured and sent for sequencing analysis (together with forward and reverse primers 5′-TGATCCTGACGACATCCTGG-3′ and CATGGAAGGAGGGGTGATGG, respectively). Results were analyzed with Finch Tv Software.

A breeding colony was established at the Yale University Animal Resources Center. Mice were kept on a 12-h light/dark cycle. Food and water were freely

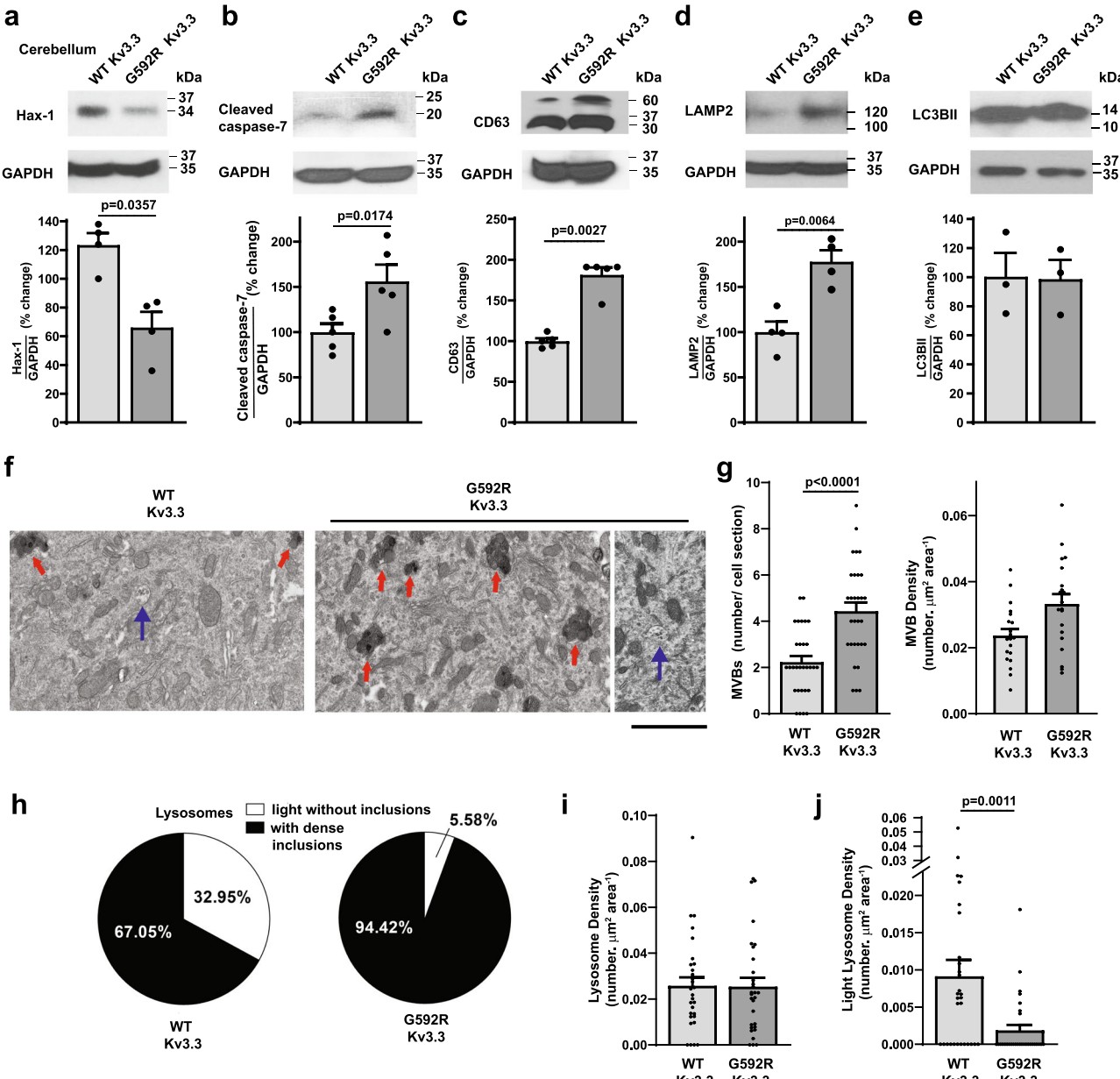

**Fig. 7 The G592R Kv3.3 mutation decreases Hax-1 and increases multivesicular bodies and lysosomes in cerebellar neurons.** Immunoblots and quantification demonstrating a decrease in Hax-1 levels (**a** $n = 4$ independent experiments; data are presented as mean ± SEM; two-tailed paired $t$ test), an increase in activated caspase 7 (**b** $n = 5$ independent experiments; data are presented as mean ± SEM; two-tailed paired t test), an increase in CD63 levels (**c** $n = 5$ independent experiments; data are presented as mean ± SEM; two-tailed paired $t$ test) and an increase in LAMP2 (**d** $n = 4$ independent experiments; data are presented as mean ± SEM; two-tailed paired $t$ test) but no change in LC3BII (**e** $n = 3$ independent experiments; data are presented as mean ± SEM; two-tailed paired $t$ test) in the cerebellum of G592R Kv3.3 mice compared to those in wild type mice. **f** Representative electron micrographs of somata of Purkinje cells in wild type and G592R Kv3.3 mice. Red arrows depict lysosomal structures with dark inclusions and blue arrows point to multivesicular bodies, scale bar 2 μm. **g** Quantification of the numbers of multivesicular bodies (MVB) detected in cross-sections of electron micrographs of somata (*left*) and of the density of MBVs in the cytoplasm (*right*) of Purkinje neurons in wild type and G592R Kv3.3 mice. **h** Representation of proportions of lysosomes either lacking (light) or containing dense inclusions in wild type and mutant mice. **i** Cytoplasmic density of total lysosomes in wild type and mutant mice. **j** Quantification of endosomes lacking dense inclusions in wild type and G592R Kv3.3 mice. The data in (**f–j**) represent measurements from 30 sections each from three wild-type and 3 G592R Kv3.3 mice (data are presented as mean ± SEM; two-tailed unpaired $t$ test). Source data and uncropped Western blots are provided as a Source Data file.

available. All experiments were done in accordance with the Yale University Institutional Animal Care.

**Cell counting**. We used the technique of fluorescence-activated sorting of nuclei from fixed tissue to count the density of neuronal nuclei in the cerebella of wild type and G592R mice[22]. After dissection, cerebella were submerged for 5 min in isopentane cooled with dry ice and then stored at −80 °C before homogenization

and purification of neuronal nuclei. Neuronal nuclei were stained with an Alexa 488 conjugated anti-NeuN antibody (Millipore Sigma, MAB377X, 1:1000) before counting by flow cytometry (BD LSRII flow cytometer).

**EEG recordings**. Local field potentials (LFPs) were recorded using a 16-channel silicon probe (A1x16; NeuroNexus Technologies, Inc, Ann Arbor, MI) from the cerebellum in wild-type and G592R mice ($N = 4$ for each group). Recordings were

**Hax-1 Wild type**

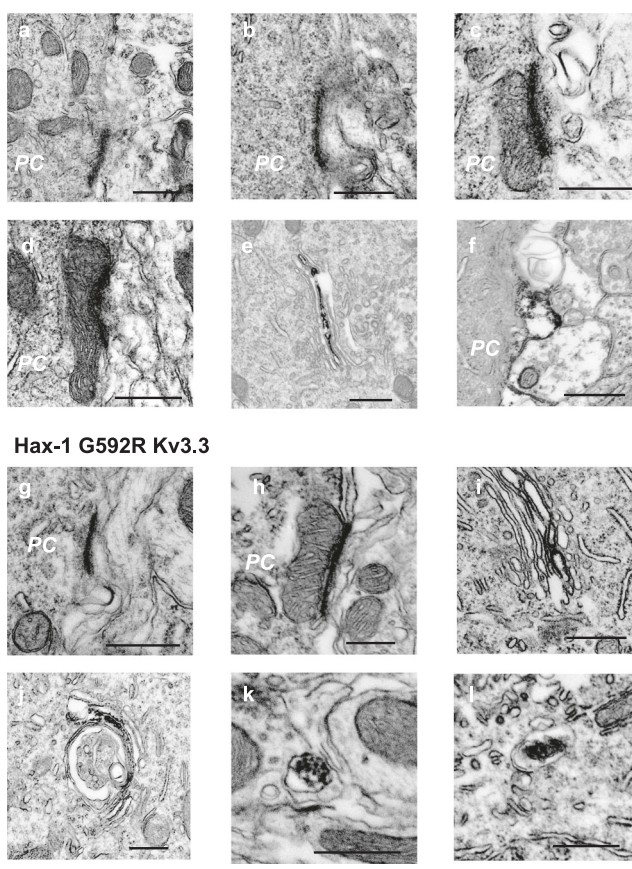

**Hax-1 G592R Kv3.3**

**Fig. 8 Electron immunomicroscopic localization of Hax-1 in Purkinje cells (PC) using diaminobenzidine labeling to localize immunoreactivity. a, b** Immunostaining detected at the plasma membrane of wild-type animals. **c, d** Hax-1 immunoreactivity at sites where mitochondria are apposed to the plasma membrane in wild-type animals. **e** Hax-1 on endoplasmic reticulum (wild-type mice). **f** Hax-1-immunoreactive protrusions at the soma of a neuron from a wild-type animal. **g** Hax-1 at the plasma membrane in G592R Kv3.3 mice. **h** Mitochondrial/plasma membrane Hax-1 immunoreactivity in G592R Kv3.3 mice. **i** and **j** Hax-1 on endoplasmic reticulum in G592R Kv3.3 mice. **k** and **l** Examples of multivesicular bodies with internal Hax-1 immunoreactivity in G592R Kv3.3 animals. Hax-1 immunostaining is representative of 40 and 32 images taken from sections of three wild type and three G592R Kv3.3 mice, respectively. Scale bars for all figures, 500 nm.

performed in 4 months old male mice anesthetized with urethane (1.5 g/kg) given intraperitoneally. After a stable plane of anesthesia was achieved, mice were placed in a stereotaxic frame (Kopf, Tujunga, CA) on a temperature-regulated heating pad (Physitemp Instruments Inc., Clifton, NJ) set to maintain body temperature at 37–38 °C and craniotomy was made above cerebellar vermis. The recording probe was positioned at midline, 6.2 mm posterior from bregma[48] and slowly lowered 1.7 mm from the surface through the cortex depth. LFPs were amplified using A-M System amplifier (model 3600, Carlsborg, WA) with filters set between 1 and 500 Hz, digitized at a rate of 1 kHz, and collected on a computer via a CED Micro1401-3 interface and Spike2 software (Cambridge Electronic Design, Cambridge, UK).

Quantitative offline power spectrum analyses were performed using Matlab (Mathworks, Natick, MA). LFPs signals were filtered with eegfilt.m from EEGLAB[49]. The magnitude of power in each frequency band (delta: 0.5-4 Hz; theta: 4–8 Hz; alpha: 8–12 Hz; beta: 12–30 Hz; gamma: 30–80 Hz; and gamma band oscillations (GBO): 80–300 Hz) was calculated as the average power in a given band across the 8 most superficial channels over 5-min long epochs for each animal. Distribution profiles of cortical high-frequency gamma oscillations power (heat maps), were derived using power of LFPs spatially smoothed by linear interpolation between values at each recording site. Statistical comparisons between wild-type and G592R mice were performed using two-tailed t-test (GraphPad Prism Software, Inc, La Jolla, CA).

**RotaRod experiments**. Rotarod test (AccuScan Instruments) was used to evaluate motor coordination in WT and G592R KV3.3 mice. Before the test, mice were trained twice per day for 3 consecutive days. Mice were placed back on the rod immediately after falling, up to 3 time per session. The rotarod was accelerated from 0 to 90 rpm over 180 sec. The day of the test, mice were placed on the Rod and latency to fall off the rod was recorded.

**Dowel walking test**. Mice were placed in the center of a suspended horizontal dowel (0.9-cm wide) and allowed to walk along the dowel towards a platform[23,24]. On reaching the platform they were returned to the center of the dowel. Performance was evaluated by measuring the number of times each mouse reached the platform during a 2-minute interval. If a mouse fell off the dowel, it was replaced at the center of the dowel, but if it fell off five times, it was discarded. In all experiments, only male mice were used, and the experiments were carried out blind to genotype.

**Western blot and Co-immunoprecipitation**. Cultured CHO cells were harvested and washed twice in cold PBS and were then resuspended in cold homogenization buffer (25 mm Tris. HCl, pH 7.4, 150 mM NaCl, 1%NP-40, 1 mM EDTA, 5% glycerol, and complete Mini Protease Inhibitor Tablet (Roche Applied Science)). They were homogenized and centrifuged at $13,000 \times g$ for 15 mins (at 4 °C) to remove large cell debris. The supernatants were then aliquoted, quickly frozen in liquid nitrogen, and stored at -70 °C until use. Protein estimation was done using Bradford's reagent (Bio-Rad). For Western blot, samples were suspended in 1× sample buffer for electrophoresis. For co-immunoprecipitation[50], samples were incubated with primary antibodies at 4 °C overnight. 100 µl of prepared protein A/G beads (Thermo-Scientific) were added to samples and rotated for 2 h at 4 °C. The beads were washed stringently six times in a homogenization buffer containing 1× Complete EDTA-free protease inhibitor cocktail tablet followed by centrifugation. 50 µl of sample buffer was added to the beads and incubated at room temperature for 30 min before loading on SDS-PAGE gel. After electrophoresis, the proteins was transferred onto polyvinylidenedifluoride membranes (Bio-Rad). Blots were then blocked in PBS containing 10% nonfat milk for 1 h at room temperature with shaking. Blots were then incubated with the respective primary antibody overnight at 4 °C. After three washes with blocking buffer, blots were incubated for 1 h with horseradish peroxidase-conjugated secondary antibodies, followed by extensive washes in PBS. Labeled proteins were detected by enhanced chemiluminescence. Polyclonal rabbit anti-Kv3.3 antibody (Alomone Labs) was used at 1:200; monoclonal rabbit anti-phospho TBK1 antibody (Cell Signaling Technology) was used at 1:500; polyclonal anti-rabbit Hax-1 antibody (Proteintech) was used at 1:200; monoclonal mouse anti-GAPDH antibody (Santa Cruz) was used at 1:1000; monoclonal rabbit anti-TBK1 antibody (Abcam) antibody was used at 1:1000; Polyclonal anti-rabbit CD63 (Santa Cruz) was used at 1:200; anti-pS6 antibody (Cell Signaling) was used at 1:1000; cleaved caspase-7 (ASP198) (Cell Signaling) was used at 1:500 dilution; mouse monoclonal anti-LAMP2 antibody (Santa Cruz) was used at 1:200; polyclonal anti-rabbit anti-LC3BII (Cell Signaling).

**Cell death and caspase activation**. Propidium iodide (PI) (5 µg/ml, Sigma) was added into the culture medium and incubated for 30 min at 37 °C. After washing with 0.01 M PBS three times. Cells were then imaged and counted under inverted fluorescence microscope. To evaluate the effects of Hax-1 overexpression on activation of caspase 7, Hax-1/pcDNA3(+) plasmid (kindly provided by Dr. Florian Basserman) and pcDNA3 vector alone were separately transfected into mutant G592R Kv3.3 cell lines using lipofectamine 2000. After 48 h transfection, protein was extracted and Western blots were carried out to determine the expression level of cleaved caspase 7 in the two groups of mutant G592R Kv3.3 and Hax-1 overexpression in mutant G592R Kv3.3.

**Kv3.3 channel C-terminal truncations**. Kv3.3 full length and C-terminal truncations Kv3.3Δ were amplified via PCR, including N –terminal HindIII restriction site and Kozak sequence GCCACC before start codon, a C-terminal 6xHis tag sequence, ATGGTGGTGGTGATGATG and Xba1 restriction site before stop codon. The two restriction sites were used for cloning the truncations into pCDNA3 vector. All constructs were confirmed by restriction digestions and sequence analysis.

**Patch-clamp recordings in CHO cells**. The patch electrodes were pulled from 1.5-mm OD borosilicate capillary glass (World Precision Instruments). The resistance of a typical electrode was 2-3 MΩ for whole-cell recording when filled with intracellular solution. For whole-cell recordings, the intracellular solution consisted of (in mM) 97.5 potassium gluconate, 32.5 KCl, 10 HEPES, 5 EGTA, pH 7.2, with KOH. The bath solution consisted of (in mM) 140 NaCl, 5.4 KCl, 1.3 CaCl₂, 25 HEPES, 33 glucose, pH 7.4, with NaOH. Series resistance was 2-4 megaohms and was compensated by 80–85%. The data were acquired at 10 kHz and filtered at 5 kHz. Data were acquired using pClamp9 software (Molecular Devices).

**Immunocytochemistry**. CHO cells expressing wild type Kv3.3 were grown on glass coverslips to ~80–90% confluence. Coverslips were then washed three times in PBS,

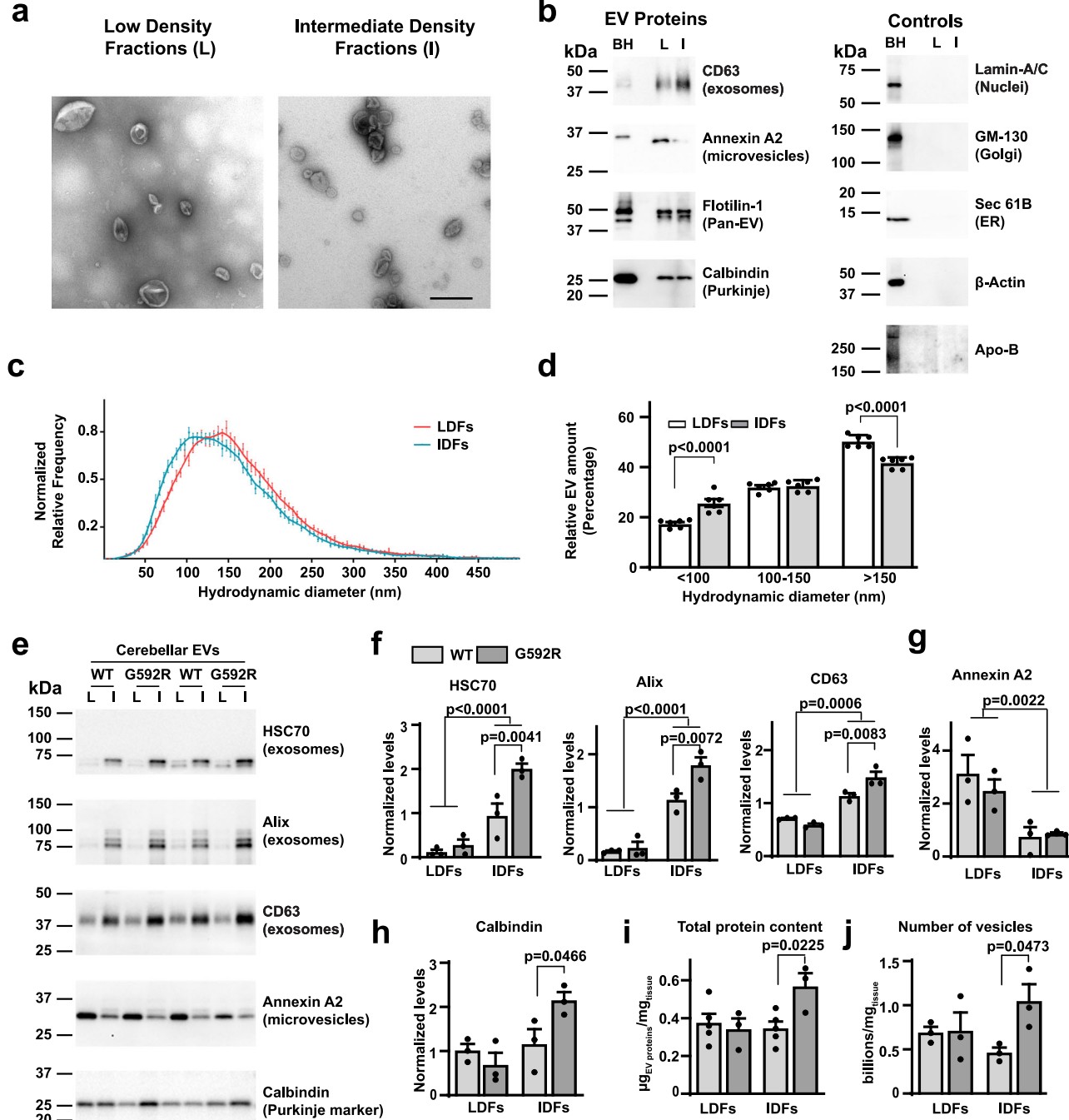

**Fig. 9 The Kv3.3 G592R mutation promotes release of exosomes into the extracellular space. a** Representative transmission electron microscopy photomicrographs of low density fraction (LDF) and intermediate-density fraction (IDF) extracellular vesicles (EVs) isolated from the cerebellum of a wild type mouse. Scale bar: 500 nm. Images are representative of ten images taken of each of the sets analyzed. **b** Representative Western blot analyses of LDF EVs (L) and IDF EVs (I) compared to the cerebellar homogenate (BH). **c** Nanotrack analysis (NTA) of the hydrodynamic diameter of the EVs found in LDFs and IDFs. The curve is normalized to the mode of each distribution (five independent isolations. **d** Quantification of particles in LDFs and IDFs by NTA that are found within the size bins shown in the graph. Data are plotted as percentage of total number of EVs. Five independent isolations. **e** Representative Western blot analyses of EVs isolated from cerebella of wild type and G592R Kv3.3 mutant mice. LDFs (L) and IDFs (I) were isolated from cerebella of two mice of the same genotype and these were combined and tested for the markers shown. Two independent experiments are shown in the blot. **f–h** quantification of data in **e**. The densitometric quantification was performed on three independent experiments, representative of six mice per genotype. BCA protein assay (**i**) and total particle count as estimated by NTA (**j**) of LDFs and IDFs in wild type and G592R Kv3.3 mice. Values were normalized to the cerebellum wet weight (five independent isolations for wild type and three independent isolations for mutant mice, corresponding to ten and six mice, respectively. All data are shown as a mean ± SEM. Statistical test used in all figures: Two-Way ANOVA with Sidak's multiple comparisons test). Source data and uncropped Western blots are provided as a Source Data file.

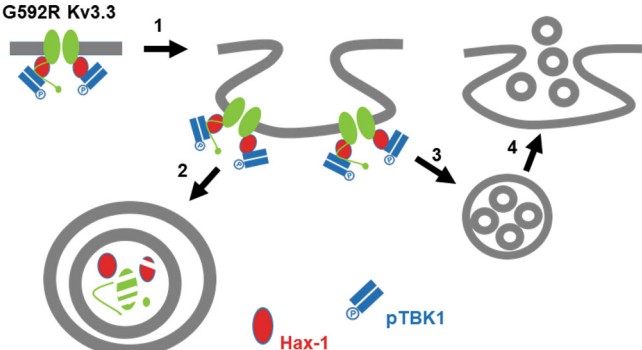

**Fig. 10 Diagram of potential interactions in G592R Kv3.3-induced TBK1 signaling.** The figure shows endocytosis of the receptor complex (1) and trafficking into multivesicular bodies (2 and 3) that can fuse with lysosomes (2) or promote the release of multivesicular bodies in the form of exosomes (4).

and then cells were fixed in freshly made PBS containing 4% paraformaldehyde for 10 min. Coverslips were washed three times with PBS, and then cells were permeabilized by incubating in a PBS solution containing 1% BSA and 0.2% Triton X-100 for 10 min. This buffer was used for all subsequent incubations. Primary antibodies were added to the coverslips, and the reaction was allowed to take place for 1 h at room temperature. Coverslips were washed and then incubated with Alexa fluor 488 goat anti-rabbit and Alexa fluor 594 goat anti-mouse antibodies for 30 min. The coverslips were then washed three times, mounted on glass slides with the Citifluor mounting medium (Ted Pella, Inc., Redding CA), and then viewed via fluorescence microscopy.

**Electron microscopy**. Mice were anaesthetized and transcardially perfused with 4% PFA and 0.1% glutaraldehyde. After post-fixation overnight, vibratome sections (50 μm) containing the cerebellum were washed and then immunostained for Kv3.3 or Hax-1 for 48 h at 4 °C using an anti-Kv3.3 antibody raised in rabbit (Alomone labs, APC-102) or an anti-Hax-1 antibody raised in rabbit (Santa Cruz Biotechnology, sc-29268). The sections were then incubated with biotinylated goat anti-rabbit secondary antibodies (Vector, 1:250) for 1.5 h at room temperature followed by incubation in Avidin biotin-peroxidase (Vector, 1:200) for 1.5 h at at room temperature. Immunoreactivity was then visualized with a diaminobenzidine (DAB)/glucose oxidase reaction[51]. The sections were then osmicated (15 min in 1% osmium tetroxide) and dehydrated in increasing ethanol concentrations. During the dehydration, 1% uranyl acetate was added to the 70% ethanol to enhance ultrastructural membrane contrast. Flat embedding in Durcupan followed dehydration. Ultrathin sections were cut on a Leica ultramicrotome, collected on Formvar-coated single-slot grids, and analyzed with a Tecnai 12 Biotwin electron microscope (FEI).

**Isolation of EVs from adult cerebella**. Cerebellar EVs were isolated using a minor modification of a technique previously used for hemibrain EVs[15]. Briefly, 2 snap-frozen cerebella per genotype (either wild type or G592R Kv3.3 mutant mice) were finely minced and treated with 20 U/mL papain (Worthington, Lakewood, NJ, US) in Hibernate A (BrainBits, Springfield, IL, US) for 15 min at 37 °C. The enzymatic digestion was stopped by the addition of ice-cold 5 μg/mL leupeptin, 5 μg/mL antipain, 5 μg/mL pepstatin, 1 mM PMSF, 1 μM E64 (all reagents from Sigma-Aldrich, St. Louis, MO, US) in Hibernate A. The solution was then centrifuged at 300 × g for 10 min at 4 °C and the supernatant filtered first through a 40 μm cell strainer (Fisher Scientific, Pittsburgh, PA, US) and then through a 0.2 μm surfactant-free cellulose acetate (SFCA) syringe filter (Corning Incorporated, Corning, NY, US). The cleared mixture was centrifuged at 2000 × g for 10 min at 4 °C and the supernatant centrifuged again at 10,000 × g for 30 min at 4 °C. The supernatant was then ultra-centrifuged at 100,000 × g for 70 min at 4 °C. The pellet containing crude EVs was washed once in PBS and resuspended in a 40% OptiPrep solution, containing 10 mM Tris-HCl pH7.4, 0.25 M sucrose, and 40% OptiPrep (all reagents from Sigma-Aldrich). Decreasing concentrations of OptiPrep (20%, 15%, 13%, 11%, 9%, 7%, 5%) were carefully layered on the top of it in order to set up a density step gradient that was centrifuged overnight at 200,000 × g at 4 °C. 1.5 mL fractions (eight fractions in total) were collected, resuspended in PBS and centrifuged again at 100,000 × g for 70 min at 4 °C. Pellets were resuspended in PBS and either lysed in 2× RIPA buffer with protease inhibitors (RIPA composition: 2% Triton X-100, 2% sodium deoxycholate, 0.2% SDS, 300 mM sodium chloride, 100 mM Tris-HCl pH 7.4, 2 mM EDTA, all reagents from Sigma-Aldrich) or loaded into a nanotrack analysis (NTA) machine (ZetaView model PMX-220 TWIN, Particle Metrix, Meerbusch, Germany) to estimate the amount and the dimension of the vesicles. A small aliquot of brain EVs was also imaged under transmission electron microscopy upon fixation with a paraformaldehyde (PFA)

solution containing 2% w/v PFA in a 100 mM sodium cacodylate buffer (reagents from Electron Microscopy Sciences, Hatfield, PA, US). The density of each fraction was calculated on the basis on the density of the buffer ((0.25 M sucrose, 10 mM Tris-HCL, pH 7.4) and the density of specific concentrations of OptiPrep provided by the manufacturer. Low density fractions (LDFs, comprising fractions 1, 2, and 3, density 1.054–1.074 g/ml) were combined, as were IDFs (fractions 4, 5, and 6 density 1.074–1.103 g/ml). Lysed EVs were separated on a 4–20% gradient precast polyacrylamide 26 gel (Bio-Rad, Hercules, CA, US) and transferred onto PVDF membranes (Immobilon, EMD Millipore, Billerica, MA, US). Immunoblotting was performed using the following primary antibodies: anti-Alix (1:1,000, #92880S, Cell Signaling Technology, Danvers, MA, US), anti-Annexin A2 (1:20,000, #ab178677, Abcam, Cambridge, UK), anti-Apo-B (1:1000, #ab20737, Abcam), anti-β-Actin (1:50,000, #3700T, Cell Signaling Technology), anti-CD63 (1:1,000, #ab217345, Abcam), anti-Calbindin D28K (1:1,000, #PA5-85669, ThermoFisher Scientific, Waltham, MA, US), anti-Flotilin1 (1:1000, #610821, BD Biosciences, San Jose, CA, US), anti-GM130 (1:5000, #610822, BD Biosciences), anti-Hax1 (1:200, #11266-1-AP, Proteintech, Rosemont, IL, US), anti-HSC70 (1:4,000, #sc-7298, Santa Cruz Biotechnology, Dallas, TX, US), anti-Kv 3.3 (1:400, #APC-102, Alomone Labs, Jerusalem, Israel), anti Lamin-A/C (1:100, #sc-376248, Santa Cruz Biotechnology), anti-Sec61B (1:1000, #14648S, Cell Signaling Technology), and anti-TBK1 (1:500, #3013, Cell Signaling Technology). The secondary antibodies were purchased from Jackson ImmunoResearch (West Grove, PA, US) and used following manufacturer's instructions. Protein bands were visualized with the iBright FL1500 imaging system (ThermoFisher Scientific) upon incubation at room temperature for 5 min with ECL (Pierce, Rockford, IL, US) or femto ECL (Pierce) when the signal was too faint. For statistical analysis, densitometry was carried out using the open source software ImageJ (National Institute of Health (NIH), Bethesda, MD, US) and normalized to cerebella weight with an accuracy of 1 mg. Three independent experiments, for a total of 6 cerebella per genotype, were analyzed. Groups were compared using a Two-way ANOVA with the Sidak's multiple comparisons test, considering as variables the genotype and the fractions tested, for each marker. Data are shown as mean ± SEM (standard error of the mean). The data shown in this paper comply in full with the minimal information for studies of EVs 2018 (MISEV2018) guidelines[38].

**Reporting summary**. Further information on research design is available in the Nature Research Reporting Summary linked to this article.

## Data availability

Data supporting the findings of this manuscript are available from the corresponding authors upon reasonable request. A reporting summary for this article is available as a supplementary information file. Source data are provided with this paper.

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

## Acknowledgements
This work was supported by NIH grants DC01919 and NS111242 (to L.K.K.), AG052005, AG051459, AG052986 and AG067329 (to T.L.H.), AG017617, AG056732, AG057517 and DA044489 (to E.F.) and a grant from the National Ataxia Foundation (to Y.Z.). We thank Drs. John Kozarich for interactions critical to the initiation of this project and Florian Bassermann for providing the Hax-1 construct. The authors thank Tal Hargash, Matthew Ríos, Cynthia Hughes, and Judith Stein for technical expertise and Dr. Chris N. Goulbourne for TEM imaging of EVs. The authors also thank Craig Kelley, who helped in EEG analyses, in particular calculation of GBO heat-maps.

## Author contributions
Y.Z. carried out the biochemical, electrophysiology, and light level immunohistochemical experiments on CHO cells and western blots on cerebellar tissue. L.V. carried out the behavioral experiments and western blots on cerebellar tissue. K.S.-B. carried out all electron microscopy experiments. A.W., J.H.M., and R.A.F. generated the mutant G592R Kv3.3 mice used in this study. M.S. carried out EEG recordings and analyses. M.S.P. carried out the fractionation and counting of neuronal nuclei. P.D'A. and E.L. carried out the isolation of exosomes from cerebellar tissue. Y.Z. and L.K.K. wrote the manuscript. L.K.K. and T.L.H. conceived and oversaw the project.

## Competing interests
The authors declare no competing interests.
