## [Peer Review File · Nature Communications]

REVIEWER COMMENTS

Reviewer #1 (Remarks to the Author):

The manuscript by Zhang et al describes a well-designed series of experiments to investigate the relationship of TBK1 in the regulation of Kv3.3 potassium channels and its potential mechanisms of neurodegeneration. Mutation in KCNC3 are known to be causative to SCA13, with degeneration in patients ranging from mild to severe. The authors describe the current background information on this particular potassium channel, including the binding with HAX1 for survival of cerebellar neurons.

The authors used Crisp/Cas 9 to generate a patient based mutation in KCNC3 in mice. The mice developed some mild functional abnormalities as displayed by reduced function of rotarod test at 4 and 7 months of age, in addition to the dowel walking test to measure coordination. The overall structure of the cerebellum revealed no gross abnormalities, however recording from PC neurons showed increased activation. Western blotting revealed no increase in expression levels between mutant and wildtype mice, however EM analysis revealed some subcellular mislocalization. The authors went on to identify potentially misregulated kinases in these mutant animals and discovered that TBK1 was upregulated in the cerebellum >2-fold and that TBK1 and Kv3.3 could be immunoprecipitated. The authors then performed a series of deletion experiments in the channel to identify the binding domains. Additional experiments determined the effects of depolarization of Kv3.3 and the resulting increase in activity of TBK1. Further experiments using inhibitors of TBK1 demonstrated the resultant decrease in HAX1 binding with Kv3.3. Further experiments in CHO cell lines supports the conclusion that mutated Kv3.3 channels traffics HAX1 to the endosomal pathway. Moving to in vivo experiments, the authors noted a 50% reduction in HAX1 in mutant animals and an increase in multivesicular bodies.

This reviewer found this manuscript to contain a well thought out series of experiments with conclusions that supported the data. The information on Kv3.3, the mutant channel, and its discoveries regrading TBK1 represent important information. The authors continued to explore in sufficient depth of mechanism associated with abnormal trafficking. This reviewer would recommend acceptance of the manuscript.

Reviewer #2 (Remarks to the Author):

Report to authors

The authors have previously reported (ref.10 – Cell 2016) that Kv3.3 potassium channels bind the anti-apoptotic protein Hax-1 and that this facilitates the formation of a stable complex of the channel's C-terminus with the actin cytoskeleton by linking to the actin-regulatory proteins Arp2/3. This complex serves to slow the decay of the Kv3.3 current by preventing N-type fast (ball-and-chain) inactivation. They also found that a Kv3.3 channel with a mutation in the C-terminus (G592R) that is associated with a form of spinal cerebellar ataxia (SCA13) also bound Hax-1 but did not permit activation of Arp2/3 and hence destabilized the channel-actin complex.

In the present paper the authors have taken knowledge of the Kv3.3 channel complex, and of the defect in SCA13, an important step further by showing that the binding of Hax-1 to Kv3.3 requires the associated binding and activation of a kinase enzyme called Tank Binding Kinase 1 (TBK1). This is required to ensure the stability of the channel – actin complex and suppress fast N-type channel inactivation. They also find that the G592R mutated channel associated with the SCA12 cerebellar ataxia overstimulates TBK1 and thereby enhances the incorporation of the survival protein Hax-1 into lysosomes and so promotes cell death.

ASSESSMENT

This is a very interesting paper of sufficient novelty and general interest to appeal to a variety of scientists beyond the specialist in Kv3.3 potassium channels.

The first point of general significance is that it adds a carefully documented example to the accumulating evidence that the ancillary proteins to an ion channel's pore subunit may not be simply devices for regulating the channel's electrical behaviour but can also provide a route to interaction with the rest of the cell as a participant in the cell's overall biology. Thus, within the potassium channel family, an early example was the beta subunit for the *Drosophila* Shaker (Kv1.2) channel, which (like the Kv3.3's Hax-1/TBK1 proteins), could be viewed simply as a device for controlling the rate of inactivation (Rettig et al., 1994: *Nature*, 369,289) but turned out to be a member of the aldo-keto reductase family of membrane proteins (Chouinard et al, 1995: *PNAS*, 92, 6763) and so acts as a reporter able to adjust the channel's (and cell's) electrical activity to changes in the cell's oxidative state and hence decide when the fruit-fly should go to sleep (Kempf et al, 2019). In the present study, the "beta-subunit" equivalents turn out to be a cell survival (anti-apoptotic) protein and a vesicular trafficking protein. To my knowledge these are unique among ion channel auxiliary subunits and would not have been predicted to serve such a function *ab initio* (at least, by me).

A second point of novelty and significance is the tracking of a plausible mechanism whereby the Kv3.3 (G592R) potassium channel mutation associated with the type 13 spinal cerebellar ataxia can also produce cerebellar degeneration. It is proposed that this arises through overstimulation of TBK1 when bound to Kv3.3 (G592R) leads to the partial dissociation and loss of the anti-apoptotic protein Hax1, coupled with accelerated down-trafficking of Kv3.3 protein complexes and other membrane proteins by excess TBK1. *Inter alia*, the authors devised a novel Kv3.3 (G592R) homozygote knock-in mouse by CRISPR/Cas9 gene-editing. In adulthood, these mice developed the symptoms of impaired motor control associated with cerebellar ataxia.

The authors' hypotheses are well-backed by substantial multi-technical experiments, the principal results of which are densely presented in the figures. Some of these (such as the generation of the Kv3.3 (G592R) knock-in mice) are highly innovative and others represent some heavy-duty work (such as testing the activity of the Kv3.3 (G592R) mutant channel on >260 different kinases, leading to the subsequent studies on TBK1).

COMMENTS.

I have no strong criticisms of either the authors' experiments or their interpretation, only a few comments and questions.

1. Why, in looking for additional subunits beyond Hax-1, did the authors specifically zero in on kinases? And how unique was the twofold increase in phosphorylated TBK1 compared with all of the other 260 kinases studied (or have to be to have convinced the authors that TBK1 was worth pursuing?)
2. Starting from a naked Kv3.3 channel, neither Hax-1 or TBK1 would seem a priori likely to form a channel subunit. Do the authors envisage any other components to the Kv3 channelosome? Thus, would Hax-1 or TBK1 emerge as likely dancing partners from a general selection procedure such as a yeast 2-hybrid screen, or might there be other suitors?
3. Did the authors not compare the Kv3.3 currents in cerebellar neurons in or from wild-type and mutant knock-in mice, and the effect of the TBK1 inhibitor thereon, or only in CHO cells? If so, how do cerebellar Kv3.3 currents compare with those generated by CHO-expressed channels? (This might have a bearing on point 2)
4. Given a choice, I would have preferred scattergrams to bar charts throughout though this might be difficult in view of the complexity of the figures (see below).
5. The manuscript is well-written and clear overall but presents a tedious read in the present format because of the complexity of the individual figures; These in turn require long figure legends (on a separate page) so one has repeatedly to switch from text description to a figure then to its legend then back to text page. I would have preferred individual figures dealing with one point or type of experiment at a time. However, I suppose such compound figures as used in

this paper are standard for this type of scientific journal but they do little for the reviewer's enjoyment or temper, or the reader's vision.

Reviewer #3 (Remarks to the Author):

The authors attempted to isolate small extracellular vesicles called exosomes from mouse cerebellum (Fig 8). Exosomes are generally isolated from extracellular fluids such as CSF, blood or cell culture fluids, not interstitial fluid in frozen tissue such as cerebellum. Isolation of exosomes from tissue can be done, however thorough validation needs to be undertaken to demonstrate successful isolation of vesicles and rule out co-isolation with microsomes, synaptic vesicles, exosome mimetics e.t.c. The authors did carry out western blotting for markers common to exosomes (these markers are not specific to exosomes), however no additional data is provided. The authors need to demonstrate that they have isolated EVs by performing electron microscopy, probing for exosome negative markers and particle sizing (refer to the International Guidelines ISEV for further details). No conclusions can be drawn from the Kv3.3 G592R mutation in regards to exosome release from the data provided.

REVIEWER COMMENTS

Reviewer #1 (Remarks to the Author):

1) *This reviewer would recommend acceptance of the manuscript.*

We sincerely thank the reviewer for their positive comments.

Reviewer #2 (Remarks to the Author): *I have no strong criticisms of either the authors' experiments or their interpretation, only a few comments and questions.*

Again, we very sincerely thank the reviewer for their helpful comments. In response to specific questions, we have made changes to the manuscript as follows:

1). *Why, in looking for additional subunits beyond Hax-1, did the authors specifically zero in on kinases? And how unique was the twofold increase in phosphorylated TBK1 compared with all of the other 260 kinases studied (or have to be to have convinced the authors that TBK1 was worth pursuing?)*

Determination of which kinases are activated by the mutation provides a clear major indicator of which signaling pathways are altered and can be more sensitive than simply looking for changes in levels of specific proteins. We found that the change in TBK1 was the most significant change we detected in the cerebellum ($p < 0.0005$, $n = 6$, cerebellum, $n = 11$ forebrain). In the same screen, changes of marginal significance were detected in only two other kinases, (Mitogen and Stress activated kinase 1 ($p = 0.033/0.2127$ t-test \pm Welch correction) and CamK1b ($p = 0.08/0.032 \pm$ Welch correction)). We have now added this information to the revised manuscript.

2) *Starting from a naked Kv3.3 channel, neither Hax-1 or TBK1 would seem a priori likely to form a channel subunit. Do the authors envisage any other components to the Kv3 channelosome? Thus, would Hax-1 or TBK1 emerge as likely dancing partners from a general selection procedure such as a yeast 2-hybrid screen, or might there be other suitors?*

Hax-1 was indeed detected as the most prominent hit in a yeast 2-hybrid screen that used the cytoplasmic C-terminal domain of Kv3.3 as bait, and we have now added this information to the introduction of the manuscript. TBK1 was not detected in that screen, suggesting that that the interaction may require additional factors (perhaps such as Hax-1 itself?). We have now added a statement to the discussion clarifying this. For the record, we also feel that there must be other dancing partners involved with the channel, but do not speculate on that here!

3) *Did the authors not compare the Kv3.3 currents in cerebellar neurons in or from wild-type and mutant knock-in mice, and the effect of the TBK1 inhibitor thereon, or only in CHO cells? If so, how do cerebellar Kv3.3 currents compare with those generated by CHO-expressed channels? (This might have a bearing on point 2)*

We have not yet compared the Kv3.3 currents in cerebellar neurons in wild-type and mutant mice. We agree that this could be a worthwhile endeavor, but the isolation of the Kv3.3 component in adult neurons is not a straightforward task, particularly because, in certain other neurons, changes in one Kv3 family member result in compensatory changes in other channels

in the same family. Thus, a thorough analysis requires comparison of mice with multiple mutation/knockout phenotypes, and would not directly relate to the activation of TBK1 by the channels. Optimally we would prefer to consider that as an independent study.

4) *Given a choice, I would have preferred scattergrams to bar charts throughout though this might be difficult in view of the complexity of the figures (see below).*

As requested, we have added scattergrams to the bar graphs in a way that we hope has not added too much complexity to the figures.

5. *The manuscript is well-written and clear overall but presents a tedious read in the present format because of the complexity of the individual figures; These in turn require long figure legends (on a separate page) so one has repeatedly to switch from text description to a figure then to its legend then back to text page. I would have preferred individual figures dealing with one point or type of experiment at a time. However, I suppose such compound figures as used in this paper are standard for this type of scientific journal but they do little for the reviewer's enjoyment or temper, or the reader's vision.*

We thank the reviewer for their comment and agree one hundred percent about the need to make the manuscript as readable as possible. We are of course limited by the word limit for the bulk of the manuscript and limits on the number of figures. With the new added experiments on the exosome we are already just over the suggested work limit. In response to this comment, we have, however, separated the visual data into the ten figures that are allowed rather than the original eight. We sincerely hope this will go some way to improving enjoyment, temper and vision for all concerned, including us.

Reviewer #3 (Remarks to the Author):

The authors attempted to isolate small extracellular vesicles called exosomes from mouse cerebellum (Fig 8). Exosomes are generally isolated from extracellular fluids such as CSF, blood or cell culture fluids, not interstitial fluid in frozen tissue such as cerebellum. Isolation of exosomes from tissue can be done, however thorough validation needs to be undertaken to demonstrate successful isolation of vesicles and rule out co-isolation with microsomes, synaptic vesicles, exosome mimetics e.t.c. The authors did carry out western blotting for markers common to exosomes (these markers are not specific to exosomes), however no additional data is provided. The authors need to demonstrate that they have isolated EVs by performing electron microscopy, probing for exosome negative markers and particle sizing (refer to the International Guidelines ISEV for further details). No conclusions can be drawn from the Kv3.3 G592R mutation in regards to exosome release from the data provided.

We thank the reviewer for this major suggestion for improving this section of the manuscript. In response to the reviewer's comments we now carried out a much more detailed characterization of the EV preparation, which follow (and expands) on the MISEV2018 guidelines. The new data are now presented in the entirely new Figure 9. For clarity, we present the MISEV2018 guidelines below in green italics (Théry et al., 2018) and provide the descriptions of our new experiments that comply with them in black text.

A. 'As a rule, both the source of EVs and the EV preparation must be described quantitatively. [...] Global quantification of EVs should be provided.'

Although we previously described in detail the protocol we used for the isolation of EVs, a global quantification of vesicles was lacking. As per MISEV2018 guidelines, several methods are eligible, among them global protein quantification and particle counts. Accordingly, we performed both methods, now included in Fig. 8i and 8j. These further reinforced and confirmed our findings. By both total protein content and NTA counts of particles, the amount of exosomes in the mutant cerebella is higher than in controls, with microvesicle levels unaffected.

B. 'At least one protein of each category 1 to 3 must be evaluated in any EV preparation (at least each time pre-analytical and/or EV isolation conditions are modified). Analysis of proteins of categories 4–5 is recommended for studies that focus on one or more EV subtypes (e.g. small EVs < 200 nm, vs larger EVs: category 4), or that have identified a functional soluble factor in EVs (category 5):

1. 'To demonstrate the presence of a lipid bilayer in the material analysed, at least one transmembrane or GPI-anchored extracellular protein must be shown.'

2. 'To demonstrate that the material analysed contains more than open cell fragments, at least one cytosolic/periplasmic protein with lipid or membrane protein-binding ability must be shown (2a). Other cytosolic proteins are more promiscuously associated with EVs and other structures and thus should be only optionally used as EV markers (2b).'

3. Purity controls include proteins found in most common co-isolated contaminants of EV preparations: depending on the source of EVs, expected contaminants from category 3a (lipoproteins and serum-derived materials), or 3b (urine), should be evaluated.

4. Proteins present in subcellular compartments other than the plasma membrane and endosomes

5. The mode of association to EVs of soluble extracellular proteins with functional activities (via a specific or promiscuous receptor? Or internal?) should be determined.

Our previous data complied with points 1 and 2. We have now included more positive and negative controls for our EV characterization that can be found in new Figures 8b and 8e and that expands MISEV2018 requirements. We demonstrate the presence of at least two transmembrane EV proteins belonging to point 1 (CD63 and Annexin A2), three cytosolic EV proteins belonging to point 2a (Alix, HSC70, Flotilin-1), one cytosolic protein for point 2b (Calbindin), Apo-B as negative control for point 3 (demonstrating absence of contaminant lipoproteins) and four negative controls for point 4 (subcellular compartments other than the plasma membrane and endosomes: Lamin-A/C for nuclei, GM-130 for Golgi, Sec61B for the endoplasmic reticulum and β -Actin for the cytoskeleton-cytosol contamination). We do not provide functional studies on soluble factors in EVs so point 5 does not apply to this manuscript.

C. Characterization of single vesicles: use two different but complementary techniques, for example:

1. Electron or atomic force microscopy

2. Single particle analyzers (not electron microscope-based)

This revised manuscript now includes a new Figure 8a, which shows TEM photomicrographs of the vesicles found in LDFs and IDFs (Point 1). No debris could be found. The vesicles imaged by TEM are compatible with EVs as they show a cup-shaped morphology which is typical for

this type of vesicles. A heavy contamination of synaptic vesicles, as indicated by the concerns of Reviewer 3, can also be excluded by size, given that synaptic vesicles are smaller than EVs (Qu et al., 2009, the diameter peaks at 40 nm and 100 nm for synaptic vesicles and small EVs, respectively). The new Figures 8c and 8d show the Nanoparticle Tracking Analysis (NTA) characterization of particle size and comply with point 2. These data further reinforce the molecular evidence that LDF are enriched in microvesicles while IDF contain mainly exosomes: we report that LDF EVs are larger than IDF EVs, mirroring the relationship between microvesicles and exosomes.

D. We now recommend that the topology of EV-associated components be assessed, that is, whether a component is luminal or on/at the surface of EVs, at least for those required for a given EV-associated function.

This last point does not apply to our study because we characterize EV levels but we do not provide any EV functional data.

In summary, we believe our new data answer in full the concerns of Reviewer 3. Specifically in response to the reviewer's request that '*The authors need to demonstrate that they have isolated EVs by performing electron microscopy, probing for exosome negative markers and particle sizing (refer to the International Guidelines ISEV for further details)*', we have now provided all of this new data in the revised manuscript.

REVIEWERS' COMMENTS

Reviewer #2 (Remarks to the Author):

The authors have satisfactorily answered my questions and made appropriate changes to the manuscript. From my viewpoint the manuscript is now suitable for publication.

Reviewer #3 (Remarks to the Author):

Well done to the authors for addressing the definition of extracellular vesicle (EV) as guided by the International Committee (ISEV). The microscopy, western blots and NTA analysis are a welcome addition.

Only one issue remains to be addressed that relates to aligning the terms, low density fractions (LDFs) "which contain mainly microvesicles identified by the marker Annexin A2" and intermediate density fractions (IDFs) "which contain exosomes" with the known density of these vesicles . The authors have used an EV isolation method and gradient different to that of Jeppesen et al, so please include the densities of each fraction and state how these were calculated.

We thank the reviewer for their very positive response to the changes we made. We have now, as requested, provided the values for the densities of the IDF and LDF fractions in the Methods section, and also clarified that density of each fraction was calculated on the basis on the density of the buffer and the density of specific concentrations of OptiPrep, as provided by the manufacturer.